# Distinct methane-dependent biogeochemical states in Arctic seafloor gas hydrate mounds

Scott A. Klasek[1,2,9], Wei-Li Hong [3,4,5,9 ✉], Marta E. Torres[6], Stella Ross[6], Katelyn Hostetler[1], Alexey Portnov[4,7], Friederike Gründger [4,8] & Frederick S. Colwell[1,6]

Archaea mediating anaerobic methane oxidation are key in preventing methane produced in marine sediments from reaching the hydrosphere; however, a complete understanding of how microbial communities in natural settings respond to changes in the flux of methane remains largely uncharacterized. We investigate microbial communities in gas hydrate-bearing seafloor mounds at Storfjordrenna, offshore Svalbard in the high Arctic, where we identify distinct methane concentration profiles that include steady-state, recently-increasing subsurface diffusive flux, and active gas seepage. Populations of anaerobic methanotrophs and sulfate-reducing bacteria were highest at the seep site, while decreased community diversity was associated with a recent increase in methane influx. Despite high methane fluxes and methanotroph doubling times estimated at 5–9 months, microbial community responses were largely synchronous with the advancement of methane into shallower sediment horizons. Together, these provide a framework for interpreting subseafloor microbial responses to methane escape in a warming Arctic Ocean.

[1] Department of Microbiology, Oregon State University, Corvallis, OR, USA. [2] Department of Botany, University of Wyoming, Laramie, WY, USA. [3] Department of Geological Sciences, Stockholm University, Stockholm, Sweden. [4] Centre for Arctic Gas Hydrate, Environment and Climate (CAGE), Department of Geosciences, UiT The Arctic University of Norway, N-9037 Tromsø, Norway. [5] Baltic Sea Centre, Stockholm University, Stockholm, Sweden. [6] College of Earth, Ocean, and Atmospheric Sciences, Oregon State University, Corvallis, OR, USA. [7] University of Texas Institute for Geophysics, Austin, TX, USA. [8] Department of Biology, Arctic Research Centre, Aarhus University, Aarhus, Denmark. [9]These authors contributed equally: Scott A. Klasek, Wei-Li Hong. ✉email: wei-li.hong@geo.su.se

Microbially generated methane in marine sediments has been estimated at $10^{13}$–$10^{14}$ g per year[1]. Microbial anaerobic methane oxidation (AOM) is responsible for consuming the majority of this methane—up to 90%[1]—before it can escape to the hydrosphere. This globally widespread[2] microbial methane filter consists of very slow-growing[3,4], currently uncultured clades of anaerobic methanotrophic archaea (ANME) and often-symbiotic sulfate-reducing bacteria (SRB). These communities thrive at sulfate-methane transitions (SMTs), sediment depths where methane is oxidized with sulfate (SR-AOM)[5]. In contrast to the large areas where SMTs occur within the sediment, at discrete locations of active methane gas release, such as pockmarks and mud volcanoes, over 90% of the methane can escape aerobic and anaerobic oxidation by benthic organisms and end up in overlying waters[6].

Methane release from the Arctic seafloor has received significant attention over the past two decades[7]. Seafloor methane venting to the hydrosphere has been documented along a wide portion of the East Siberian Margin[8], the South Kara Sea shelf[9], and the upper slope of the Beaufort Sea[10]. Extensive geophysical surveys have characterized thousands of fault-associated seeps below warming waters along the West Spitsbergen (Svalbard) margin[11,12]; numerical modeling and U/Th dates from authigenic carbonates revealed that seepage has persisted here for hundreds to thousands of years[13,14].

The Storfjordrenna trough mouth fan, ~50 km south of Svalbard, hosts gas hydrate-bearing mounds (GHMs) on the seafloor that are morphologically similar to submarine pingos described in the Beaufort[15] and Kara[16] Seas (Fig. 1). These GHMs lie below water depths of 370–390 m, which approach the upper limit of gas hydrate stability in this area[17]. Gas hydrates within these sediments are thus sensitive to changes in oceanographic conditions and particularly susceptible to Arctic Ocean warming. Gas leakage was observed above four of five GHMs, which are thought to have formed from hydrate accumulation and methane gas overpressure following glacial retreat[17]. Microbial community responses to subsurface methane release, whether driven by tectonic[18], climate[19], and/or oceanographic[20] forcing, are important to constrain because they support macrofaunal communities[21] of ecological and economic importance[22]. However, how these microbial communities respond to changes in methane release over time in Arctic cold seeps remains largely uncharacterized. As environmental changes from either natural or anthropogenic causes could potentially result in increased methane flux, placing the responses of sediment microbial communities in a temporal context is of immediate importance.

In the Arctic Ocean, abrupt release of methane from gas hydrate dissolution in the central Barents Sea has been hypothesized[23], while methane release from the Deepwater Horizon oil spill into deep Gulf of Mexico waters was correlated with the growth of aerobic methane-oxidizing *Gammaproteobacteria* and oxygen drawdown[24]. Sediment microbial community responses to fluctuating methane states have been characterized at mud volcanoes[25,26], and methane has recently been found to shape community structure at Storfjordrenna GHMs[27]. However, a dynamic understanding of how microbial activity may mitigate methane release in methane-rich marine sediments is currently poorly understood.

Changes in concentration gradients of porewater sulfate in marine sediments have been used to constrain the timing of submarine landslides[28], to infer rates and fluxes of sulfur through sulfate-reducing bacterial communities[29], and to indicate irrigation (through bioturbation or ascending gas bubbles[30]) or migration of upwards-diffusing methane[14]. Under steady-state conditions with a constant methane flux, sulfate concentrations decrease linearly with depth until the SMT is reached[31], assuming all SR is coupled to AOM. In contrast, sulfate profiles at locations experiencing increases in methane flux change to a concave-up shape, as sulfate concentrations decrease abruptly to <1 mM over tens of cm (Fig. 2).

Reactive transport modeling of this transition from linearity towards a concave up shape in porewater sulfate profiles (Fig. 2, Fig. S1) can be used to estimate how long ago methane began to diffuse into shallower sediment zones, provided that other phenomena (advection, seawater irrigation, bioturbation, or mass transport deposits) are minimized or constrained[28]. This scenario attributes the thinning of the sulfate reduction (SR) zone to methane advancement into shallower sediment layers, which

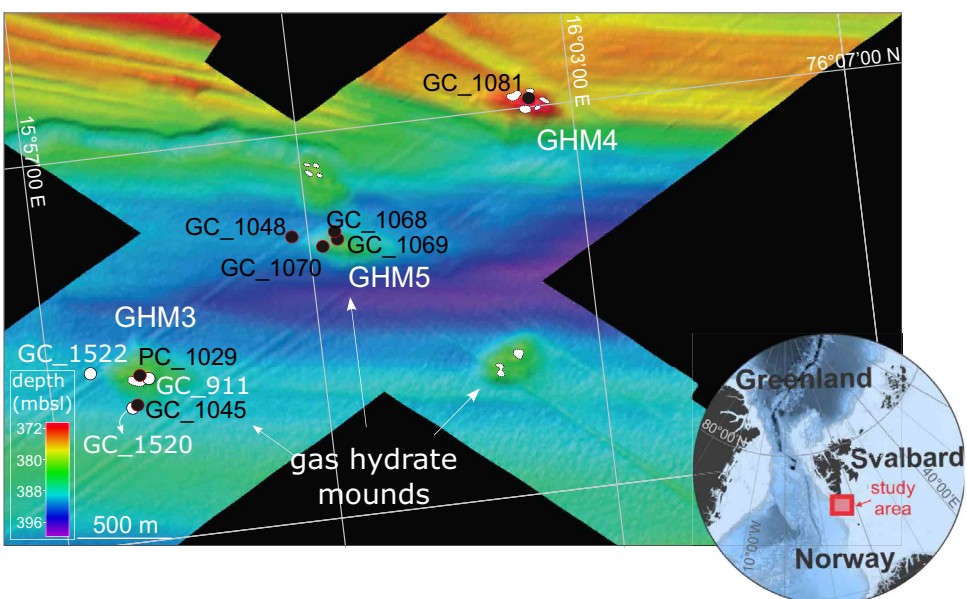

**Fig. 1 Bathymetric map of Storfjordrenna gas hydrate mounds and core locations.** Storfjordrenna is located south of the Svalbard Archipelago in the red box on the regional map. GHM: gas hydrate mound, GC: gravity core, PC: push core, mbsl: meters below sea level. Black points show cores collected and first described in this study, while white points indicate cores described in previous studies (see Table S3 for a summary of porewater data available from Storfjordrenna). White polygons at GHMs indicate areas of seafloor gas release observed at the time of the cruise in 2016.

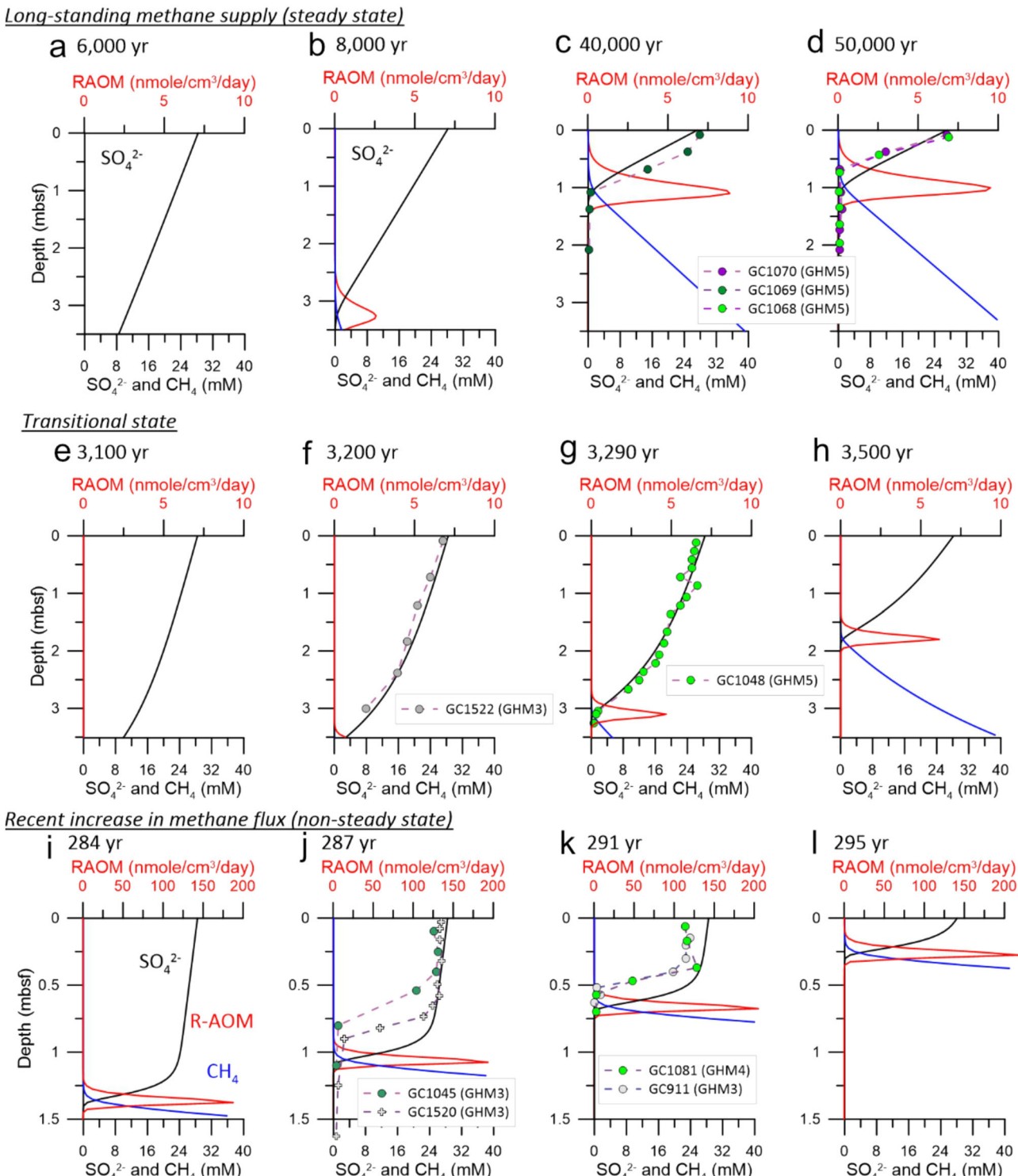

**Fig. 2 Time-progressing reactive-transport modeling of porewater geochemistry under three states of methane dynamics.** States include a long-standing steady-state methane supply (**a–d**), a transitional state (**e–h**), and a recent increase in methane flux (i–l). Porewater sulfate concentrations are shown as black lines, methane concentrations as blue lines, and rates of AOM (RAOM) as red lines. Data from gravity cores (GCs) are shown for the upper few meters below seafloor (mbsf). Porewater data from GHMs representing different methane states (GHMs 3 & 4, recent methane flux increase; GHM5, steady state) were shown for comparison, with colored points representing cores first described in this study and gray points representing previously described cores (see Fig. 1 for site locations). **a–d** With a long-standing methane supply, the porewater sulfate profile approaches a steady state after 40 kyr. The SMT gradually shoals with time (at a rate of 7E-3 cm/yr when comparing **b** and **c**) with a linear decrease in downcore sulfate concentration resembling the profiles obtained from three gravity cores recovered at GHM5. **e–h** A transitional state occurs when the increase in methane supply is only moderate. **i–l** For a system experiencing a recent increase in methane supply, AOM stimulated by a recent pulse of methane rapidly changes sulfate concentration gradients as observed in sediment cores recovered from GHMs 3 & 4.

stimulates AOM within them. This framework has been used to characterize Storfjordrenna GHMs[14] in combination with observations of free gas, gas hydrates, and other geochemical signatures to support a model where episodic methane emission occurs in pulses, with distinctive pre- and post-active stages[32].

In this work, we constrain temporal responses of microbial communities as methane migrates upwards towards shallow sediment horizons. Using samples and data from Storfjordrenna GHMs, where varying states of methane transport are evident, we employ geochemical, numerical, and molecular approaches to report shifts in rates of AOM, abundances of ANME and SRB, and microbial community patterns concomitant with recent changes in methane flux. These analyses reveal a tightly coupled microbial response to intensifying subseafloor methane flux at a prime location in the Arctic Ocean where gas hydrate is susceptible to ocean warming.

## Results

**Field descriptions and general patterns.** Black-colored glacio-marine sediments were recovered in all cores, reflecting the precipitation of iron sulfide minerals resulting from high rates of sulfide production[33]. Authigenic carbonate nodules were retrieved in several cores, and chunks of gas hydrates several cm in diameter were observed between 40–50 cm below seafloor in a replicate of push core (PC) 1029. Cores PC1029 and GC1081 were taken from areas of gas seepage indicated by the white polygons in Fig. 1. Core recovery lengths ranging from 102 to 335 cm captured SMTs in all cores except for PC1029 (Table S1). All cores show downcore increases in alkalinity throughout the sulfate reduction zone, providing further support of AOM as the dominant sink for sulfate (Figs. 3a, 4a, and 5a). In situ methane concentrations are probably higher than those reported, as gas samples were taken from cores at atmospheric pressure. No bubbles or frothy sediment texture was observed in the recovered cores, limiting the possibility of degassing upon core retrieval.

Bacterial and archaeal 16S rRNA gene sequencing recovered 3.12 million sequences and 16,470 amplicon sequence variants (ASVs) after contaminants were removed (see methods). Bubble plots (Figs. 3b, 4b, and 5b, left panels) show the fifteen most abundant taxonomic classes in the dataset, each of which individually constitute 1% or more of the total sequences, and combined account for 83.6% of reads in the dataset. The three most common ASVs, which alone comprise 22.2% of all sequences, belong to the class JS1 (phylum *Atribacteria*) which are thought to ferment organic matter[34]. Two other dominant classes, *Deltaproteobacteria* and *Methanomicrobia*, are subdivided into families of anaerobic methanotrophs (ANME) and genera of sulfate-reducing bacteria (SRB), respectively (Figs. 3b, 4b, and 5b, right panels). Respectively, ANME and SRB make up 10 and 12% of total sequences in this dataset, and ANME are most dominant at or near SMTs. Two clades of sulfate-reducing bacteria that commonly associate with ANME at seeps, SEEP-SRB1 and SEEP-SRB2[35,36], share similar distribution patterns. Droplet digital PCR counts of the methane-fixing methyl-coenzyme reductase gene *mcrA* and dissimilatory sulfate reduction gene *dsrAB* span several orders of magnitude across cores and depths (Figs. 3c, 4c, and 5c).

**Identification of distinct states of methane transport.** Porewater sulfate profiles from the seven cores investigated in this study suggest contrasting methane dynamics and AOM rates as revealed by numerical reactive-transport modeling with a reduced reaction network[14] (Fig. 2). Thorough descriptions of the modeling approach are available in the Methods section and Supplemental Information. Briefly, we assumed AOM as the only reaction responsible for consuming sulfate based on low porewater ammonium concentrations across several cores from

Storfjordrenna (Fig. S2, Table S3). Without fitting any porewater data, cores were classified based on model runtime from an initial condition after which the shape of the modeled sulfate profile roughly matched the observed concentrations. After a runtime of 40 kya, the model approaches steady state with an SMT at ~1 m below seafloor (Fig. 2c–d).

Three cores showing linearly decreasing sulfate concentrations with depth (GC1068, GC1069, and GC1070) have profiles consistent with a long-standing steady-state methane supply over tens of thousands of years (Fig. 2a–d). We hereafter refer to these three cores as "steady-state", though the sparsity of sulfate concentration data for these cores adds some uncertainty to this interpretation. Porewater, microbial community, and functional gene abundance data for these cores are shown in Fig. 3. In addition to these three cores, GC1048 and GC1522 are offset from GHMs and represent a special case. Linearly decreasing sulfate concentrations above 2.5 mbsf suggest a weak but persistent methane supply, but below, steeper decreases in sulfate concentration appear to reflect a recent change in methane flux consistent with moderate AOM rates and SMT shoaling speeds of 0.4 cm/yr (Fig. 2e–h). We consider these cores as belonging to a separate transitional state. Data from GC1048 are shown in Fig. S3.

In contrast, two cores with abrupt changes in sulfate concentration gradients (GC1045 and GC1081) are experiencing a recent increase in methane flux that was initiated less than three centuries ago, agreeing with previously described observations of cores GC911 and GC1520[14]. We thus consider these cores as "non-steady-state". The increase in methane supply shoals the SMT by 10 cm/yr, and numerically derived AOM rates from these two cores are an order of magnitude higher than the cores from the former groups (Fig. 2a–h). Data from these non-steady-state cores are shown in Fig. 4.

At a seep site atop GHM3, where persistent hydroacoustic gas flares over multi-year surveys detail active methane seepage[17], downcore changes from a remote-operated vehicle (ROV)-guided push core (PC1029) capture biogeochemical signatures that reflect high methane flux, gas bubble emission, and/or bioturbation (Fig. 5). The likelihood of advective fluid movement here prohibits classification with our diffusion-based modeling scheme, so we hereafter consider this seep site as a distinct state of methane transport.

**A steady-state pore fluid system.** Three gravity cores from GHM5 showed approximately linear decreases in sulfate, with methane present only below the SMT (Figs. 2a–d and 3a). Sulfide profiles track the shape of the alkalinity curves, peaking at SMT depths. In addition, macroscopic SMT-associated mucoid biofilms consisting predominantly of ANME-1[37], were observed in a split core at 63 and 68 cm in GC1070 (Fig. 3a). For these cores, we estimate depth-integrated methane fluxes of 1.3 mol m$^{-2}$ yr$^{-1}$ (Table S1) and peak rates of AOM at 10 nmol cm$^{-3}$ day$^{-1}$ (Fig. 2d). Though ANME-1a and ANME-1b each comprise 4.5% of reads across all samples from this study, ANME-1a are more abundant than ANME-1b in steady-state cores (Fig. 3b). In GC1068, *mcrA* counts above 10$^6$ copies per gram are seen just above the SMT, though gene abundance profiles otherwise display considerable variability and *dsrAB* counts are typically low, below 10$^5$ copies per gram bulk sediment (Fig. 3c).

**Non-steady-state sites showing increasing methane flux.** GC1045 was sampled from the southern margin of GHM3, and GC1081 from the center of GHM4 (Fig. 1). Sulfate profiles from these cores show concave-up curvature, suggesting that the methane-sulfate dynamics are not at steady state, but likely reflect a recent increase in methane flux[14] (Figs. 2i–l and 4a). Porewater

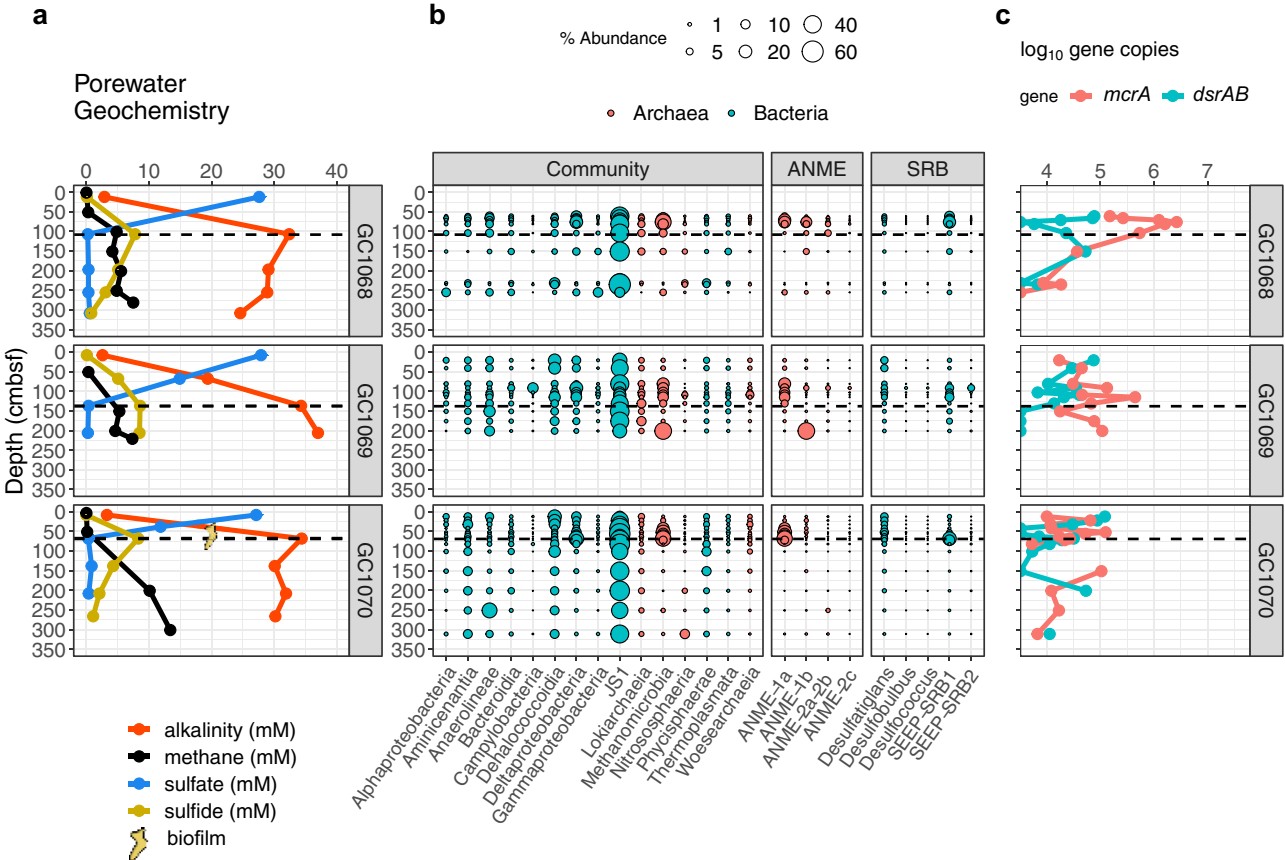

**Fig. 3 Geochemical, microbial community, and gene abundance data from three cores at gas hydrate mound 5 showing steady-state sulfate-methane dynamics.** Sulfate-methane transition depths in centimeters below seafloor are indicated by dashed lines. **a** shows methane concentrations and porewater sulfate, sulfide, and alkalinity, and **b** indicates percent abundances of dominant bacterial and archaeal classes, dominant anaerobic methanotrophic archaeal (ANME) families, and sulfate-reducing bacterial (SRB) genera. **c** shows copy numbers of *mcrA* and *dsrAB* genes per gram bulk sediment, with values below the detectable limit ($10^3 \, \mathrm{g}^{-1}$) along the margin of the panel. Macroscopic translucent-to-yellow biofilms, shown as yellow symbols in panel (**a**), were observed at 63 and 68 cm below seafloor in gravity core GC1070 (symbol size not to scale with depth axis).

sulfate profiles show a rapid decrease in concentration down core and SMTs are well established. Our modeling estimates that total methane fluxes throughout these two cores have increased over the past two decades (Table S2). Modeling scenarios were constructed on a prior dataset of several porewater species from Storfjordrenna in an attempt to account for other processes, including advection, but only a scenario applying contrasts in methane flux adequately fit the observed sulfate, ammonium, iron, and calcium profiles[14]. No fractures, mass transport deposits, porosity changes, or evidence of bioturbation were found in the gravity cores analyzed, and a buildup of ammonium to 60 μM in the first 50 cm of GC1045 (Fig. S2a, Table S3) allows us to discount the possibility of oxic bottom water intrusion. Fluxes are integrated from all modeled AOM rates, assuming AOM as the only sink for sulfate (see Supplementary Material and Fig. S2 for justification). Following these constraints, our model estimates peak AOM rates for an increasing methane flux scenario at ~200 nmol $\mathrm{cm}^{-3} \, \mathrm{d}^{-1}$ (Fig. 4d), over an order of magnitude higher than those derived for steady-state cores (Fig. 2).

In GC1045 and GC1081, percent abundances of *Deltaproteobacteria* and *Methanomicrobia* are 12% and 5.9% higher than in steady-state cores, respectively, and ANME-1b are the most abundant ANME genus (Fig. 4b). Counts of *mcrA* reach maxima around $10^7$ copies per gram at SMTs in both cores (Fig. 4c). Higher *dsrAB* abundances at shallower depths in GC1045 likely reflect a larger or more diverse sulfate-reducing community than in GC1081.

**Active methane seepage.** PC1029 was recovered from an established patch of frenulate siboglinid tubeworms (*Oligobrachia sp.* CPL clade, Fig. S4) whose chemosynthetic lifestyles are supported by sulfide generated from SR-AOM at sites with high methane discharge[21,38,39]. Observations of vigorous gas bubbling and recovery of gas hydrate support the inference that the site was experiencing high methane seepage at the time of sampling. Sulfate concentrations at near-seawater values up to 10 cm below seafloor at PC1029 (Fig. 5a) may be attributed to seawater infiltration (siboglinid bioirrigation or bubble-driven convection) and/or sulfide oxidation from bacterial symbionts[40]. Further downcore, the incomplete drawdown of sulfate and high methane concentrations suggest that sulfate-coupled AOM is an ongoing process, pointing towards a high methane flux at the center of GHM3. As processes other than sulfate diffusion from seawater are not accounted for in our model parameterization, we are unable to precisely calculate AOM rates from PC1029. Our rough estimation of the AOM rate based on the part of the sulfate profile with the greatest concentration gradient (10–15 cmbsf) yields a peak AOM rate on the order of $10^3$ nmol $\mathrm{m}^{-3} \, \mathrm{d}^{-1}$. This rate estimate would be increased significantly by accounting for siboglinid-driven pumping of bottom seawater sulfate, or sulfide reoxidation mediated by their endosymbionts. Nevertheless, this estimated AOM rate is an order of magnitude higher than the rates calculated for cores experiencing increases in methane flux shown in Fig. 2e–h.

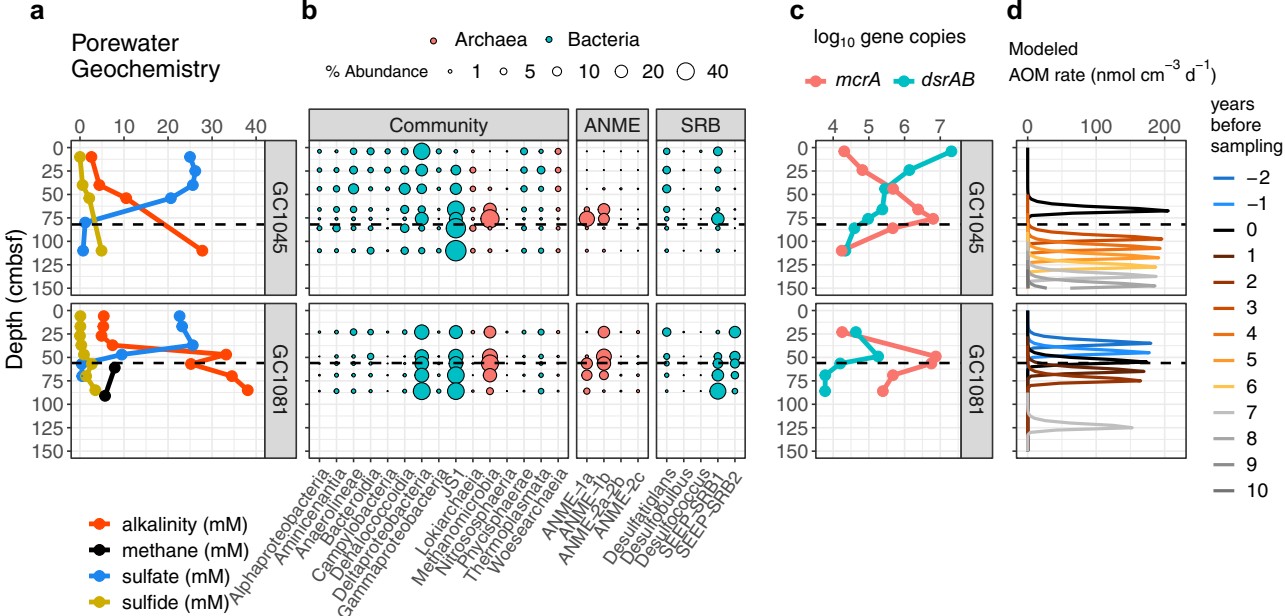

**Fig. 4 Geochemical, microbial community, and gene abundance data from two sites showing sulfate-methane dynamics suggestive of recent increases in methane flux.** Gravity cores GC1045 and GC1081 are located at gas hydrate mounds 3 and 4, respectively. Sulfate-methane transition depths in centimeters below seafloor are indicated by dashed lines. **a** shows methane concentrations and porewater sulfate, sulfide, and alkalinity, and **b** indicates percent abundances of dominant bacterial and archaeal classes, dominant anaerobic methanotrophic archaeal (ANME) families, and sulfate-reducing bacterial (SRB) genera. **c** Copy numbers of *mcrA* and *dsrAB* genes per gram bulk sediment. **d** Temporal progression of modeled AOM rates from 10 years before sampling to up to 2 years after sampling.

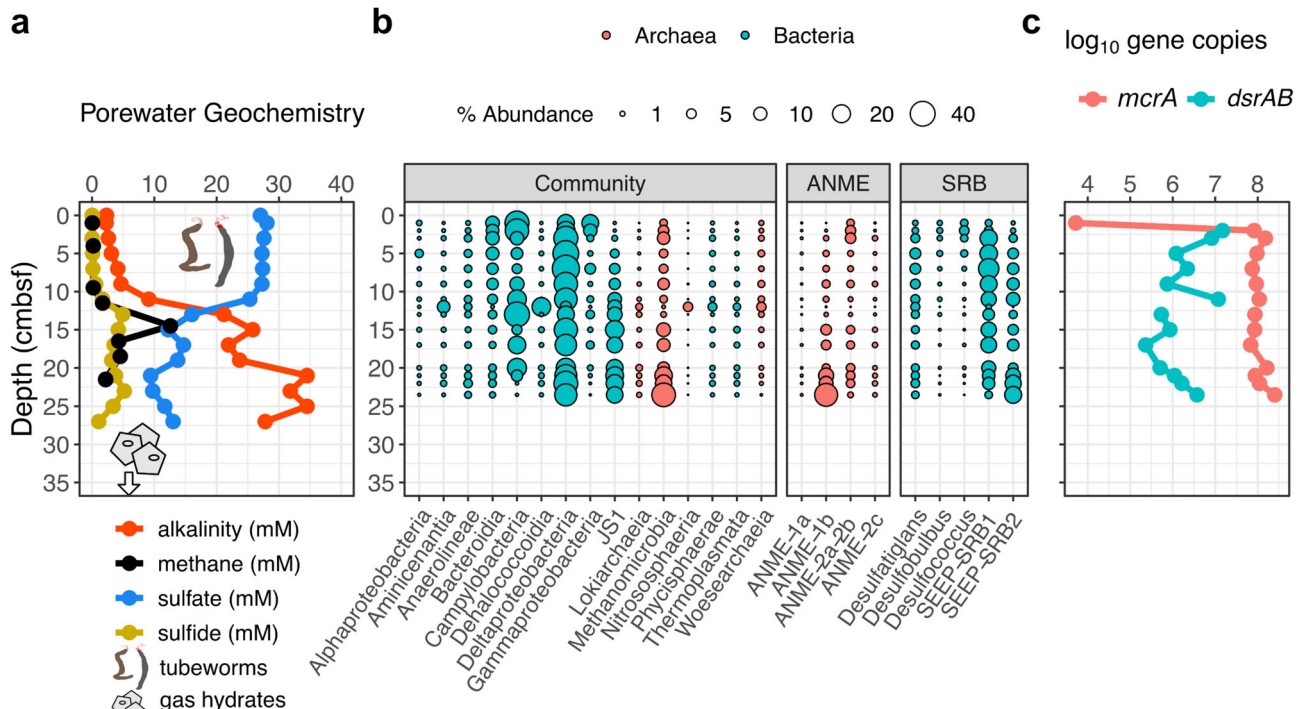

**Fig. 5 Geochemical, microbial community, and gene abundance data from an active seep site.** Push core PC1029 is located at the seep in the center of gas hydrate mound 3. **a** shows methane concentrations and porewater sulfate, sulfide, and alkalinity concentrations with depth (cmbsf, centimeters below seafloor) in addition to frenulate siboglinid tubeworms and gas hydrate nodules several cm in diameter recovered in a replicate core at 40–50 cmbsf. **b** depicts percent abundances of dominant bacterial and archaeal classes within the microbial community (left panel), dominant anaerobic methanotrophic archaeal (ANME) families (center panel), and sulfate-reducing bacterial (SRB) genera (right panel). **c** shows log₁₀ copy numbers of *mcrA* and *dsrAB* genes per gram bulk sediment.

Microbial communities from PC1029 show higher percent abundances of several classes, notably *Bacteroidia* and *Gammaproteobacteria* (at 5.4 and 3.4%, respectively), than in cores representing other states of methane dynamics (Fig. 5b). ANME-2 are the dominant ANME type at 1–3 cm in PC1029, but ANME-1b predominate at depths with lower sulfate concentrations (Fig. 5b). ANME-1a are nearly absent, agreeing with recent observations from this seep location[27] and contrasting with the two other states. Near-equal abundances of SEEP-SRB1 and SEEP-SRB2 at PC1029 are reminiscent of GC1081, the other core from a seep site (Figs. 4b and 5b). The highest *mcrA* concentrations, exceeding $10^8$ copies per gram bulk sediment, were recovered in PC1029, even in depths with high sulfate, low methane, and low alkalinity (Fig. 5c); these values are comparable to ANME cell counts reported from other seep sites[25,41]. Counts of *dsrAB* were over an order of magnitude lower than *mcrA* throughout the core, but still higher than those in nearly all other samples from different methane states.

**Response times of ANME and SRB to methane pulses as inferred from porewater modeling.** In non-steady-state cores, modeled AOM peaks migrate upward with time (Fig. 4d). Running the model backwards or forward in time reveals an upward migration of SMT at a linear rate of 10 cm per year given the bottom methane flux we assigned (Fig. 4d, Table S2). Different depths, and thus microbial communities therein, can be assigned by the time they experienced (or are expected to experience) this upward-migrating AOM peak. GC1045 communities from 66, 76, 86, and 110 cm depths, respectively, correspond to AOM peaks at the time of collection, and one, two, and over four years before. The highest concentrations of *mcrA* and relative abundances of SEEP-SRB1 and total ANME are seen in the community sampled at 76 cm (Fig. 4c, d), suggesting these taxa dominate microbial communities after about a year following methane migration into this sediment horizon. In contrast, relative abundances of ANME-1b are highest in the community from 66 cm, which may reflect a quicker growth or a preference for lower methane concentrations compared to ANME-1a. In the timespan from one to four years after the AOM pulse has passed through, *mcrA* abundances decreased by nearly three orders of magnitude, but *dsrAB* by less than one. After four years, the AOM pulse moves onward and microbial communities are starved of sulfate, ANME and SRB populations respectively decrease from 46% to 1.1 and 22% to 1.8% of the total community.

GC1081 communities from 56.5, 69, and 86 cm correspond to maximum AOM rates from the time of sampling, one and a half, and three years ago, respectively, while the community at 49 cm is associated with high (but not yet peaking) AOM rates (Fig. 4b, d). In contrast to communities from GC1045, ANME percent abundances do not decrease as quickly, and SEEP-SRB1 increases with depth (Fig. 4b). A similar trend of ANME-1b growth preceding ANME-1a is noticed, but surprisingly ANME-1b are present in high relative abundance at 24 cm, where AOM rates are not expected to be significant until two years after sampling. Concentrations of *mcrA* and *dsrAB* both roughly correspond to the present-day AOM pulse, showing no lag time with respect to methane influx (Fig. 4c). In communities from both steady-state and non-steady-state cores, *mcrA* gene abundances correlate positively with rates when plotted on a log–log scale (Fig. S5).

**Microbial community diversity and analysis.** The three most abundant classes in our dataset, the *Methanomicrobia*, *Deltaproteobacteria*, and JS1, a class of *Atribacteria*, are especially dominant in communities from cores experiencing recent methane influx (Fig. 4b). Besides these major groups, other poorly

understood taxa include the *Aminicenantes*, *Anaerolineae*, and *Phycisphaerae*, all thought to be fermentative saccharolytic heterotrophs[42–44]. *Dehalococcoidia*, also abundant, contain members capable of reductive dehalogenation[45]. We identified 76 ASVs in our dataset whose relative abundances were significantly different across communities when grouped according to states of methane dynamics (Fig. S6). These ASVs on average comprise 17.4% of the sequences in communities associated with active seepage, 1.6% of communities experiencing methane flux, and 6.9% of steady-state communities. When compared to the other two states, communities from sites displaying steady-state sulfate-methane dynamics contained higher abundances of several ASVs belonging to *Aminicenantia*, *Dehalococcoidia*, and *Woesearchaeota* (Fig. S6). In addition, one ANME-1a ASV was higher in this group, though only four of the 41 ANME ASVs in the entire dataset were differentially abundant across methane states. Several ASVs belonging to SEEP-SRB1 and *Desulfatiglans* also displayed variation among states (Fig. S6).

In communities from cores experiencing high methane flux (seep and non-steady-state cores), Shannon-Weiner alpha diversity indices decrease as depths approach peak model-derived AOM rates (Fig. 6a). Linear regressions show no such decrease in diversity across AOM peaks from steady-state areas (Fig. 6b). Interestingly, in samples from core GC1069, the highest diversity is seen at depths of peak AOM, while the opposite is apparent in GC1070 (Fig. 6b).

Differences in community structure are evident across states of methane dynamics, with communities from PC1029 showing particularly clear separation from those in steady-state cores (weighted Unifrac, Fig. 7a). These distinctions were still observed even when seep samples were omitted (PERMANOVA $R^2 = 0.087$, $p < 0.001$), and when samples from above or below the SMT were considered separately ($R^2 = 0.29$, $p = 0.001$; $R^2 = 0.18$, $p = 0.004$, respectively). In addition, we classify samples according to three geochemical zones they inhabit based on the shapes of porewater sulfate profiles: the linear sulfate reduction (SR) zone, the nonlinear SR zone impacted by recent methane influx, and below-SMT. Community structure also varied significantly across these redox zones, though PERMANOVA tests revealed only 11.6% of the variance among communities could be explained by redox zone (Fig. 7b) in contrast to 25% by methane state. Though containing high relative abundances of ANME, communities in nonlinear SR regions of non-steady-state cores were more similar to below-SMT communities than those in linear SR zones, suggestive of recent adaptations to methane influx (Fig. S7). Aside from methane states and redox zones, communities also varied significantly by the GHM and core they were sampled from (37.2% and 24.1% of variance), suggesting these Arctic GHM communities contain a high degree of biogeographic heterogeneity[27] that remains unconstrained.

## Discussion

The presence of distinct states of methane flux at Storfjordrenna GHMs allows us to examine concomitant changes in inferred AOM activity and microbial community composition. We conceptually summarize results from integrated geochemical, numerical, and microbiological analyses that characterize three distinct biogeochemical states corresponding to changes in methane supply across six cores (Fig. 8).

In Fig. 8a, steady-state sulfate and methane profiles are observed when methane consumption is balanced by sulfate diffusion from seawater. Abundances of ANME and SRB often do not peak around the SMT, and these populations are accompanied by several other microbial groups (Fig. S6), many of which

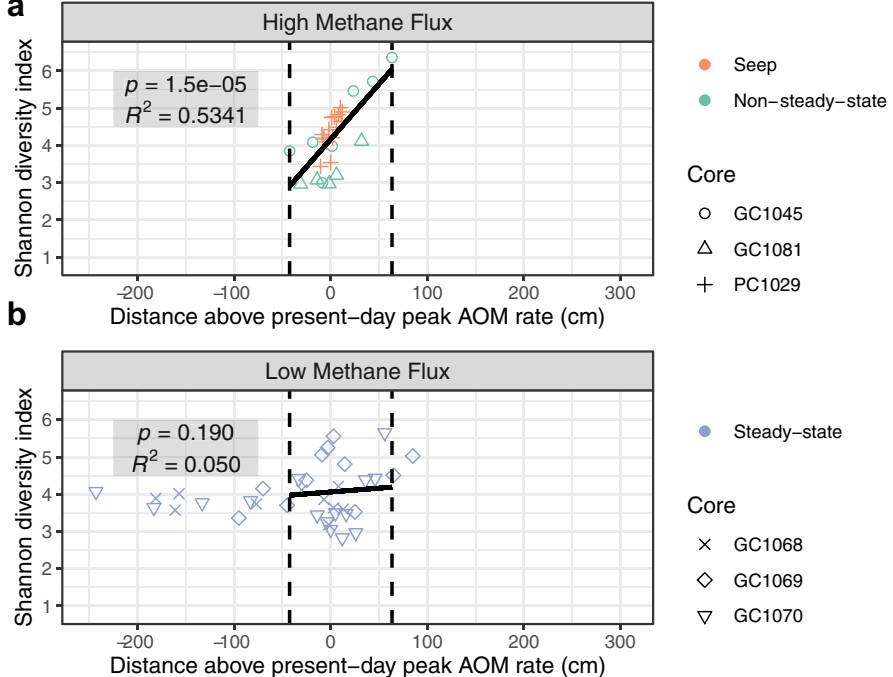

**Fig. 6 Microbial community diversity patterns across depths above or below peak modeled rates of AOM.** Shannon diversity indices of microbial communities for individual samples plotted by their distance above (positive) or below (negative) the depths corresponding to present-day maximum AOM rates across all cores. Cores are divided by panel based on whether methane flux is **a** high, or **b** low, and colored according to different states of methane dynamics. Dashed vertical lines show the distance interval corresponding to high methane flux samples. Multiple $R^2$ and slope $p$-values are shown for linear regressions of points within these intervals. For PC1029, we assigned 13 cm as the depth corresponding to the peak AOM rate based on the steepest decrease in porewater sulfate concentration.

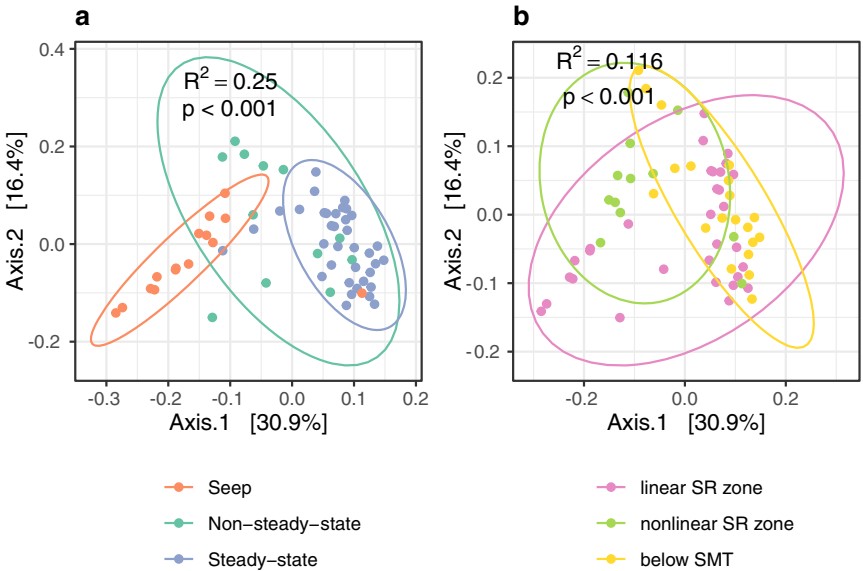

**Fig. 7 Beta-diversity of communities from gas hydrate mounds.** Principal Coordinates Analysis (PCoA) ordination of weighted Unifrac distances between all communities, colored according to states of methane dynamics (**a**), and redox zones within the sediment column (**b**). A Hellinger transformation was applied to ASV count tables before calculating the distance matrix. PERMANOVA tests verify distinct community structures present across states and redox zones, with $R^2$ and $p$-values shown in corresponding plots. SR sulfate reduction, SMT sulfate-methane transition.

may be slow-growing anaerobic fermenters of organic matter. Growth of biofilms at SMTs may also be supported over long timescales, if given steady supplies of methane and sulfate. In Fig. 8b, recent methane influx into shallower sediment horizons stimulates AOM and consumes sulfate, shoaling the SMT, and the diffusion of sulfate from seawater cannot balance the upward flux

of methane. Rates of AOM are approximately an order of magnitude higher than the steady-state cores (Fig. 2), supporting the growth of ANME/SRB and decreasing microbial community diversity. In Fig. 8c, gas seepage and the presence of hydrates at PC1029 indicate methane is at or above saturation in porewaters throughout the core. Sulfate is delivered into the sediment

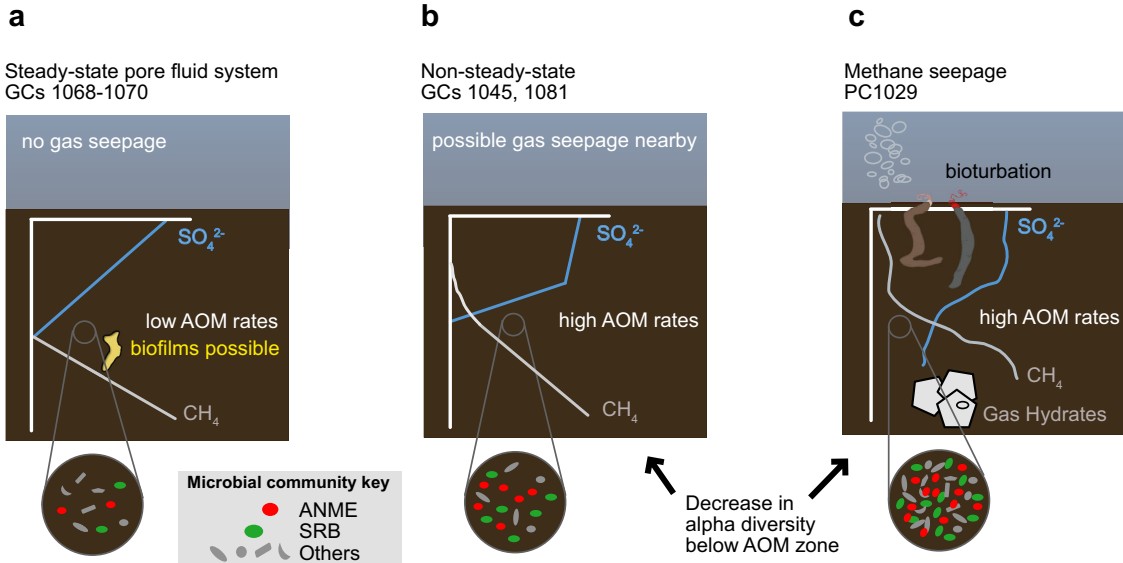

**Fig. 8 Conceptual depiction of microbial community changes concurrent with distinct states of methane dynamics at Storfjordrenna gas hydrate mounds.** Panels **a**, **b**, and **c** illustrate data shown in Figs. 3–5, respectively. Methane and sulfate profiles are shown in gray and blue lines, respectively, with microbial community changes indicated by blowup circles. ANME and SRB represent red and green ovals, with other bacteria and archaea in gray (shapes representing the diversity of other taxa). **a** Steady-state sulfate profiles suggest a weak methane influx, and low rates of AOM are observed at the SMT. Stable conditions may allow for higher microbial diversity in sediment communities and the growth of macroscopic biofilms. **b** A concave-up bend in porewater sulfate suggests recent methane migration into the sulfate reduction zone consistent with a pulse of methane beginning hundreds of years ago (approx. 290 years ago at GC1045, Fig. 2f, g). Methane travels upward throughout the sediment column, and ANME/SRB growth follows with less than a year of lag time, driving down alpha diversity. **c** Methane seepage stimulates high rates of AOM and densities of ANME/SRB. Sulfide fluxes from AOM-coupled sulfate reduction are sufficient to support frenulate siboglinid tubeworms, which may distribute sulfate across the upper several cm of sediment, increase redox gradients in underlying sediments, and further ANME/SRB growth.

column through seawater infiltration driven by bubble-driven convection, bioirrigation by frenulate siboglinids, or reoxidation of sulfide by their endosymbionts. High sulfate concentrations predict high (though unconstrained) AOM rates, supporting large populations of ANME and SRB based on respective counts of *mcrA* and *dsrAB*. The presence of several other abundant bacterial and archaeal classes suggests these shallow sediments support high cell densities overall. Though gas advects during seepage, solute transport in surrounding porewaters may remain governed by diffusion, as this decoupling of fluid transport mechanisms has been described at Storfjordrenna methane seeps[32].

Model-derived methane fluxes from Storfjordrenna non-steady-state cores GC1045 and 1081 are an order of magnitude higher than those from seeps associated with pockmark features at Vestnessa Ridge, west of Svalbard[46]. When compared to other estimates across continental margins worldwide, the magnitudes of methane flux we report for these two cores are high but well within reported ranges, while fluxes from steady-state cores are average[47]. At the seep site, PC1029, our methane flux estimate on the order of $10^2$ mol m$^{-2}$ yr$^{-1}$ is several times the maximum of other modeled AOM rates at seep sites[47], but less than the highest empirically measured AOM rate[6]. We acknowledge that the rate estimated from PC1029 is associated with large uncertainties, as we were not able to satisfactorily fit the modeled curve to empirically derived sulfate data under the current setup of the model. The model currently does not consider gaseous phase transport or bioturbation, which would enhance gaseous methane transport from deeper sediments, nor does it include sulfate infiltration from the bottom water or sulfide oxidation, which may provide additional substrates for SR-AOM. Though the timing of seepage at the center of GHM3 is unconstrained, the large populations of anaerobic methanotrophs and sulfate

reducers supported by high methane fluxes may indicate stable conditions over timescales of years[48].

We now consider ANME doubling times at sites experiencing an increase in methane flux. Though we do not have direct measures of cell activities, if we interpret the downcore increases in *mcrA* concentrations approaching the SMT (Fig. 4c) as methane-fueled ANME growth, doubling times of 147 days (GC1081, 23 to 56.5 cm) to 261 days (GC1045, 66 to 76 cm) can be derived by assuming one copy of *mcrA* per ANME genome[49]. These values complement the only other published estimate of in situ ANME doubling time, at approximately 100–200 days[48]. We can then estimate per-cell AOM rates across SMTs at an average of 0.65 pmol CH$_4$ d$^{-1}$, within range of the 0.5–1.8 reported in a bioreactor where AOM was stimulated[50]. The observation that AOM rates and *mcrA* abundances in areas experiencing increasing methane flux peak at nearly the same depths suggests that the notoriously slow ANME doubling times may not present a significant lag in the response of the benthic biogeochemical methane filter. (Considering a shoaling of 10 cm/yr in the AOM rate profile as seen in Fig. 4d, a peak *mcrA* concentration 10 cm below the concurrent AOM peak in GC1045 suggests a methanotrophic community lag time of approximately 1 year, while no lag is seen in GC1081.) These estimates contrast with the 2–5 years for ANME to become dominant and active in recently extruded subseafloor mud flows from Håkon Mosby Mud Volcano[48]. To our knowledge, this portrayal of microbial dynamics within a reactive transport modeling context is a novel approach for the methane seep literature that could be applied to other systems.

At Storfjordrenna GHMs, ANME-1 is the most abundant anaerobic methanotroph in nearly all communities (Figs. 3b, 4b, and 5b). However, our observations of ANME-2 at sulfate-rich surface sediments in PC1029 (Fig. 5b) agree with previous

findings[5]. At all other depths and locations, the reasons for ANME-1 dominance at Storfjordrenna GHMs is unclear. Genomic explanations may include the lack of an energetically expensive *nifDHK* nitrogenase in ANME-1[51], and fewer multi-heme cytochromes thought to be involved in direct intercellular electron transfer[52]. In non-steady-state cores, ANME-1a and ANME-1b were present at near-equal abundances, with ANME-1b sequences more abundant at shallower depths. In contrast, the 1a subclade was more dominant in three of four steady-state cores (Figs. 3b and 4b). Higher ANME percent abundances and *mcrA* concentrations in non-steady-state cores (Figs. 3–5) may point towards a boom-and-bust cycle where methane influx into shallower sediment layers quickly stimulates a large but ultimately unsustainable methanotrophic population, which may decline as sulfate is drawn down or as other community members establish.

In several instances, high abundances of ANME or concentrations of *mcrA* are seen at depths above those where methane is expected in GC1081 and GC1048 (Fig. 4 and Fig. S3) or below the SMT (GC1069, Fig. 3). These may reflect the limited resolution of our alkalinity and sulfate measurements. At GC1068 and GC1069, any cessation in the methane supply would allow sulfate to diffuse into a deeper depth without affecting the linearity of the sulfate profile. Alternatively, this may indicate inactive relic communities, though ANME-1 may still be capable of AOM[53] or even methanogenesis[54] when starved of sulfate.

Co-occurrences between ANME-1b and SEEP-SRB2 have been reported[2,55], and their relative abundances appear to mirror each other in GHM4 samples (Fig. 4b). Both clades of SEEP-SRB, as well as *Desulfatiglans*, are presumed to oxidize a wide variety of reduced hydrocarbons[36]. The presence of several potentially fermentative and saccharolytic clades like the *Atribacteria*, *Aminicenantes*, *Anaerolineae*, and *Phycisphaerae* may reflect alternate organic matter-dependent metabolic strategies that are interrupted by ANME and SRB when methane enters sulfate-rich porewaters. Macroscopic ANME-dominated biofilms found at two SMTs in GC1048 and GC1070[37] contained *mcrA* in concentrations of up to $7.6 \times 10^{10}$ g$^{-1}$. These biofilms may reflect sediment habitats experiencing steady methane and sulfate supply over many years (as in GC1070), or a slightly increasing methane supply (GC1048). ANME biofilms have been described at SMTs in other subseafloor locations, often in fracture-dominated cores[56].

Microbial communities inhabiting Storfjordrenna GHMs show lower richness and evenness than most other reported communities from methane seeps, sulfate-methane transition zones, and marine subsurface environments[2]. Broadly, diversity decreases with depth, but only significantly across depths corresponding to peak AOM rates in high methane flux areas (Fig. 6). In communities recently impacted by methane flux, this decrease in diversity and convergence towards a community type found below SMTs may be associated with certain taxa being outcompeted by ANME/SRB on timescales of years as the methane front progresses. Cell generation times can decrease by several orders of magnitude across the SMT, below which community assembly can be influenced more by the slow growth of a few taxa (such as *Atribacteria*) capable of thriving in an energy-limited environment as opposed to evolutionary adaptation during burial[57].

Below-SMT communities are dominated by *Atribacteria* of the JS1 class (Figs. 4b and 5b), while similar observations have been reported in methane-rich deep Antarctic marine sediments[58] and in a submarine mud volcano offshore Japan[59]. Three JS1 ASVs were identified across different states of methane flux and positions above or below the SMT (Fig. S6), though interestingly, one of them (ASV91) was preferentially abundant in above-SMT steady-state and below-SMT increasing-flux communities,

evidence towards its persistence during methane migration into shallower sediment horizons. Despite steady-state communities showing higher numbers of differentially abundant ASVs, two *Calditrichia* (genus *Caldithrix*) were more abundant in communities experiencing increased methane flux, while four *Campylobacteria* from the genera *Sulfurimonas* and *Sulfurovum* and seven *Gammaproteobacteria* were associated with active methane seepage (Fig. S6). *Sulfurovum* is capable of oxidizing elemental sulfur or thiosulfate using oxygen or nitrate as electron acceptors[60].

The presence of differentially abundant ASVs at distinct states may reflect the sampling of comparatively shallow sediments at PC1029, and an influence from macrofauna. Sulfur-oxidizing gammaproteobacterial symbionts of the siboglinid frenulate *Oligobrachia* have previously been reported in cold seeps[61]. Notably, there is an absence of *Oligobrachia* and a decreased prevalence of seafloor bacterial mats at GHM5, where steady-state cores were collected[21]. Despite the short (several km) distances between individual GHMs, many interdependent factors, such as physical disturbances, differences in fluid flow states, and colonization of foundation species provide heterogeneity across seep ecosystems[62].

In summary, our integrated approach allows us to detail states of methane transport where (A) steady-state sulfate-methane dynamics supports moderate rates of AOM at SMTs, low ANME/SRB populations, and a diverse community of organic matter degraders; (B) as methane flux increases, diffusion of methane into shallower sediment horizons stimulates ANME growth therein with lag times of a year or less, reducing community diversity overall; (C) seepage and sulfate transport into shallow sediments support high populations of ANME and SRB. Cold seeps are dynamic systems that undergo temporal perturbations in methane flux. These results highlight the importance of framing microbial community data and estimates of their metabolic processes within a spatially and temporally constrained geochemical context to more thoroughly understand microbial contributions in structuring habitats and mediating biogeochemical cycles.

The incorporation of genomic data into reactive transport models describing other microbially mediated processes has demonstrated utility in predicting subsurface microbial responses[63]. A modeling scenario that considers the dynamics of ANME growth may be of use in constraining estimates of marine subsurface methane flux into the hydrosphere. Global microbial methane filter efficiencies of 50–60%[5] have been used in modeling studies[19], but seep sites display wide heterogeneity[6]. Our finding that *mcrA* gene copy numbers correlate positively with modeled AOM rates provides some justification for coupling these populations and their associated activities (Fig. S5), mirroring the coupling of methane fluxes and transcripts of methane cyclers in peat soils[64]. Though microbial community data can provide explanatory power for predicting ecosystem processes, community changes do not always coincide with processes being measured[65]. At higher resolutions, -omics strategies capable of characterizing functional genes, transcripts, pathways, and draft genomes can link sequence data with processes and characterize ecosystem changes[66], or even apply these data into biogeochemical models to infer the presence of cryptic cycles[67]. Further studies could apply the framework discussed here towards interpreting the biogeochemistry of seep ecosystems at other locations, or to other microbially mediated cycles constrained by distinct mechanisms of solute transport.

## Methods

**Fieldwork and sample collection**. Samples and data were collected aboard the RV *Helmer Hanssen* on CAGE cruise 16–5, from June 16$^{th}$ to July 4$^{th}$, 2016, offshore Svalbard (Norway) in accordance with local laws. Bathymetric data were acquired with the RV *Helmer Hanssen*'s shipboard Kongsberg Simrad EM 302 multibeam

echo sounder using a maximum frequency of 34 KHz and a maximum swath as a function of depth of 5.5. Gas flares were detected with single (split)-beam EK60 and multibeam EM302 echosounders using 18 and 38 KHz transducers.

Gravity core (GC) 1045 was recovered from the south slope of GHM3, and GCs 1068–1070 from three locations at GHM5. GC1081 was collected near a gas seepage area at GHM4. Data from previously reported gravity cores were included in Fig. 2 and mentioned in the text (see Table s3 in Supplementary Material). Once recovered, the plastic liner containing the core was removed from the barrel, sectioned into 1 m segments, labeled, and split in half with a table saw to obtain working and archive halves. Core halves were stored horizontally at 4 °C. Following sectioning, Rhizons were used to sample porewater on archive halves. Sediment headspace gas samples for methane measurements were collected from depressurized cores, and thus should be considered underestimates for in situ concentrations. 5 ml bulk sediment was collected with cutoff plastic syringes from the working half of the core, transferred to 20 ml headspace glass vials with 5 ml 1 M NaOH and 2 glass beads, capped with rubber septa and aluminum crimpers, and stored at 2 °C. Total alkalinity (TA) was titrated onboard less than a few hours after the syringes were detached from the Rhizons. Depending on the expected TA, we used 0.1 to 0.5 ml of porewater for titration in an open beaker with constant stirring. pH was manually recorded with every addition of 0.0012 M HCl. 7–10 measurements were performed for every sample. TA was calculated from the recorded pH and amount of acid added using the Gran function[68]. Increases in porewater alkalinity determined by onboard titrations were used to roughly constrain the SMT depths (within 30 cm) for sampling purposes.

Sediment microbiology samples of 2 cm depth were then taken every 5–10 cm near the SMT and every 20–50 cm above and below it. Less than 12 h after cores were collected, ethanol-sanitized spatulas were used to scrape away the outer several mm of sediment from the working core half, and ~100 g from the interior of each sample was placed into a sterile Whirlpak bag (VWR) and immediately frozen at −80 °C.

Replicate PVC push cores for geochemical and microbiological sampling were collected ~30 cm from the seep at GHM3 using a Sperre Subfighter 30k remotely operated vehicle (ROV) equipped with a raptor arm from the Centre for Autonomous Marine Operations and Systems (AMOS). Recovery ranged from 23 to 50 cm. Rhizons were used to extract porewater from one core, and microbiology samples were extruded on deck from the other in 2-cm sections using an ethanol-sanitized spatula. These were placed into sterile bags and frozen immediately at −80 °C. Deep-frozen sediment samples were shipped from UiT-Tromsø to Oregon State University (OSU) in a Cryo Solutions MVE Doble 47 dry shipper and were subsequently stored at −80 °C.

**Geochemistry**. Sulfate content in porewater was analyzed by a Dionex ICS1100 ion chromatography (IC) at the Geological Survey of Norway (NGU). An IonPac AS23 column was equipped on the IC with the eluent (4.5 mM $NaCO_3$ and 0.8 mM $NaHCO_3$) flow set to be 1 mL/min. Due to a dilution issue when analyzing sulfate concentrations with IC, measured values were corrected by assuming a constant chloride concentration of 556 mM across the samples. From our previous knowledge of chloride concentration in the region, the concentration can be at most 10% apart from the concentration we assigned for correction (this translates to a few mM uncertainty in the sulfate concentration). However, this correction does not affect our interpretation of methane dynamics based on sulfate profiles, as we observed concomitant increases in alkalinity from TA measurements.

Total sulfide (ΣHS) concentrations were measured spectrophotometrically following the Cline method[69]. Samples were preserved onboard with 23.8 mM $Zn(OAc)_2$ solution onboard <30 min after the syringes were disconnected from the Rhizons. The samples were then kept frozen until shore-based analysis. Details of the analyses were also given in Hong et al.[68]. Depending on the factor of dilution, the detection limit is around tens of μM. To determine the concentration of dissolved methane in the porewater of the sediment matrix, a conventional headspace method was applied[17].

Gas measurements were performed using a Thermo Scientific Trace 1310 gas chromatograph equipped with a flame ionization detector (GC-FID) and a Thermo Scientific TG-BOND alumina ($Na_2SO_4$) column (30 m × 0.53 mm × 10 μm).

**Modeling**. We applied a transport-reaction model with a reduced reaction network considering only sulfate and methane, and AOM as the only reaction consuming both constituents. We simulated a 60-meter sediment column, which is the bottom of the gas hydrate stability zone in the area. AOM rates were controlled by the lower boundary condition of methane. We assigned seawater sulfate and methane concentrations for the initial and upper boundary conditions; a no-flux lower boundary condition was used for sulfate. Three different lower boundary conditions of methane were assigned to simulate the contrast states of methane dynamics as shown in Fig. 2. A higher concentration of methane for the lower boundary condition results in a more abrupt change in the sulfate concentration gradient, and thus the concave up sulfate profiles as observed from cores experiencing increased methane fluxes. On the other hand, a lower concentration of methane for the lower boundary condition in the model results in a linear decrease of sulfate concentration downcore, which resembles the profiles observed from the cores at a steady state. Additional details and assumptions are provided in Supplementary Information.

**DNA extraction, amplification, sequencing, and analysis**. DNA was extracted from sediments in a clean laminar flow hood using a Qiagen DNeasy PowerSoil kit following the manufacturer's protocol. The Earth Microbiome Project 16S Illumina Protocol was used to prepare amplicons for sequencing. Briefly, V4 regions of bacterial and archaeal 16 s rRNA genes were amplified in triplicate 25 ul reactions using universal 515-forward and 806-reverse primers[70] modified with dual-indexed Illumina sequencing adapters[71]. The thermal cycling protocol of Caporaso et al.[70] was followed without modifications. After confirming amplification with agarose gel electrophoresis, triplicate PCR products were pooled and purified with a Qiagen QIAquick PCR purification kit. Amplicon concentrations were quantified with a Qubit fluorometer using the Qubit dsDNA high sensitivity assay kit and pooled in equimolar amounts. Illumina Miseq V2 paired-end 250 bp sequencing was performed by technicians at Oregon State University's Center for Genome Research and Biocomputing (CGRB). Two sediment-free DNA extraction blanks were amplified and included in the sequencing run.

Working in R version 3.6.1, 16S rRNA amplicon data was processed with DADA2[72] (version 1.12.1) following an established pipeline[73]. Reads were denoised, chimeras removed, and taxonomies classified using version 132 of the SILVA nonredundant 16S reference database[74]. Sequences were aligned with DECIPHER[75] (version 2.12), and a phylogenetic tree was constructed using phangorn[76] (version 2.5.5). Phyloseq[77] (version 1.28.0) was used to combine read count data with sample and taxonomy information. Sequences identified as Eukaryotes, Chloroplasts, or Mitochondria were removed, and the "combined" method of decontam[78] (version 1.4.0) was then used to identify and remove 81 contaminant ASVs. In addition, after noticing the presence of *Micrococcus* in one blank sample, all four ASVs from this genus were manually removed. In total, the removed ASVs comprised 1.05% of the reads in the dataset.

Blanks and other samples with less than 8931 reads were removed, and alpha diversity metrics (ASV richness, Chao1, Shannon, and Simpson indices) were then determined. Using vegan[79] (version 2.5–6), weighted Unifrac[80] distances calculated from a Hellinger-transformed ASV count table, and PERMANOVA tests were run to assess differences in community structure among groups. DESeq2[81] (version 1.24.0) was used to identify differentially abundant ASVs among three discrete states of methane dynamics. Each above-SMT methane state was compared against the other two combined. Below-SMT samples only included two states, because all active seepage samples from PC1029 had sulfate concentrations above 1 mM. In this core, where AOM rates could only be roughly estimated, we used a peak AOM rate depth of 13 cm, which corresponded to the steepest decline in porewater sulfate.

**Droplet digital PCR**. Droplet digital PCR (ddPCR) was used to quantify abundances of functional genes *dsrAB* and *mcrA* using primer pairs described by Kondo[82] and Luton[83], respectively. Reactions of 22 ul volume were prepared in a clean PCR hood in 96-well plates using 1x Bio-Rad QX200 ddPCR EvaGreen Supermix, 200 nM primers, and 0.88 ul of tenfold-diluted genomic DNA. Droplets were generated on a QX200 AutoDG Droplet Generator using automated droplet generation oil for EvaGreen Supermix (Bio-Rad). Thermal cycling was performed immediately afterwards on a Veriti 96-well thermal cycler. Protocols began with a single initialization step at 95 °C for 5 min and then proceeded to 40 cycles of denaturation at 95 °C for 30 s, annealing for 1 min (at a temperature of 53 for *mcrA* and 58 for *dsrAB*), and for *mcrA* only, an extension at 72 °C for 75 s. Signal stabilization steps (4 °C for 5 min, then 90 °C for 5 min) were then performed before maintaining a 4 °C hold. To ensure uniform heating of all droplets, the ramp rate for all amplification cycles was set to 2 °C/minute. Reactions were kept at 4 °C overnight and read with the Bio-Rad QX200 Droplet Reader the following morning. Droplet generation and reading were performed by the lead author at OSU's CGRB core facility. Normalization was performed by inspecting fluorescence distributions using Quantasoft software (Bio-Rad). Threshold fluorescence values were manually imposed by visually inspecting distributions of DNA extraction blank and no-template-added control samples. Amplicon copy numbers per well were then converted to copies per gram wet sediment.

## Data availability

Raw 16S rRNA sequence data generated in this study have been deposited to the freely and publicly available NCBI Sequence Read Archive under BioProject accession code PRJNA533183. Geochemical, numerical modeling, and ddPCR gene count data are freely and publicly available at https://github.com/sklasek/svalflux/tree/master/data.

## Code availability

Codes used for analysis generated in this study (including phyloseq objects) are freely and publicly available at https://sklasek.github.io/svalflux/ and are archived under the DOI here: https://doi.org/10.5281/zenodo.5347747.

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

## Acknowledgements

We thank the officers and crew of R/V Helmer Hanssen on the CAGE cruise 16–5, cruise leader Michael Carroll, chief engineers Bjørn Runar Olsen, Pedro De La Torre, Frode Volden, and researcher Stein Nornæs from the Centre for Autonomous Marine Operations and Systems (AMOS) and other members of the science party for helping with sample acquisition and ROV operation. We also thank Stefan Bünz for providing bathymetric data acquired by the Norwegian Centre of Excellence, Centre for Arctic gas hydrates, Environment, and Climate (CAGE). Mark Dasenko, Anne-Marie Girard Pohjanpelto, and Jessica Nixon at the Oregon State University Center for Genome Resources and Biocomputing (CGRB) provided support with DNA sequencing and droplet digital PCR. Arunima Sen gave helpful interpretations of tubeworm observations and behavior. This work is supported by the Research Council of Norway (RCN) through its Centres of Excellence funding scheme for the Centre for Gas Hydrate, Environment, and Climate 223259, and also by the U.S. Department of Energy (DE-FE0013531). S.R. was supported by the US National Science Foundation Research Experience for Undergraduate (REU) program. M.E.T. acknowledges a fellowship from the Hanse-Wissenschaftskolleg (HWK), Germany. W.H. acknowledges the support from the RCN-funded project NORCRUST (grant no. 255150) and from the project ArcticSGD through the Norway grants and the EEA grants (UMO-2019/34/H/ST10/00645).

## Author contributions

S.A.K. and W.H. designed the study, participated in fieldwork, generated data, analyzed results, and wrote the manuscript. F.G. participated in fieldwork and contributed methane measurements. A.P. participated in fieldwork and provided bathymetric mapping. S.R. and K.H. assisted with lab work. M.E.T. and F.S.C. analyzed and interpreted results and edited the manuscript.

## Funding

## Competing interests

The authors declare no competing interests.
