## [Peer Review File · Nature Communications]

Distinct methane-dependent biogeochemical states in Arctic seafloor gas hydrate moundsReviewers' comments:

Reviewer #1 (Remarks to the Author):

This paper provides an assessment of methane dynamics in Arctic sediment environments, some of which are impacted to varying degree by methane seepage. Benthic methane fluxes in the Arctic are changing and it is important to understand how microbial communities respond to changes in fluxes. In particular, a key question involves documenting whether anaerobic methane oxidizing communities can keep pace with accelerating methane fluxes. The present paper addresses this important question.

I found this paper to be interesting and well written overall. But I do have some questions regarding a number of the fundamental assumptions and components of the data set. For example, how do the PIs know that no sulfate reduction is fueled by (non-methane) organic matter in the sediments. This is an assumption and it was not validated. Moreover, the authors acknowledge that methane concentrations may be lower than in situ because of degassing during sample collection. However, they ignore the fact that pore fluid profiles could also be altered during degassing. This needs to be mentioned and addressed. The modeling shows variable methane oxidation rates that are not at all apparent in the pore water profiles and the factors driving these patterns must be assessed.

I recommend a major revision for the manuscript and feel the paper *may be* acceptable for publication after major revision. I wish the authors all the best in revising the manuscript; I hope the suggestions below are helpful.

Specific comments (suggested insertions/changes are underlined)

Line 16: replace 'performing' with mediating

Line 43: clarify surveying (replace with ...geophysical surveys have characterized...)

Line 74: This sentence needs clarification. The findings of Kessler et al. are correlative, not causative. It remains unclear whether the oxygen drawdown was due (solely) to methane consumption vs. oil consumption. Notably, Kessler et al. did not quantify methane oxidation rates, they used oxygen as a proxy for methane and that is potentially problematic. Others have suggested that methane was not consumed efficiently during the oil spill.

Line 79: replace 'is able to' with may

Line 93: Can you pls add to the legend how far (distance) GC1048 is from the other sites?

Line 100: replace 'stages in seepage' with seepage regimes (and define what that means)

Line 115: Reference core GC1048 is presented as a low flux, non-GHM situation but in figure S1, the present-day methane flux at this site is comparable (higher but no error bars are given) to the other sites. Given this, how can it be a 'control'?

Figure 1: Sulfide and DIC concentrations are not stoichiometric to what would be expected from AOM and there is more DIC than would be predicted from sulfate drawdown if CH₄ is the sink for sulfate. In fact, sulfate drawdown is about 20 mM but there is over 30 mM DIC at the bottom of the core. It would be useful to compare drawdown of sulfate and production of DIC in relevant horizons to further support the argument that DIC is from AOM. Further, the gene data suggests that AOM related organisms are present throughout the sediment column and show no correlation with the SMT.

Line 146: I suggest using units for AOM that are more standard and general (i.e. nmol or $\mu\text{mol cm}^{-3} \text{d}^{-1}$ instead of $\mu\text{mol L}^{-1} \text{d}^{-1}$) so that the reader can more easily compare. And, yes, it is a very easy conversion but why make the reader work?

Line 164: Incorrect units – $8 \text{ mM L}^{-1} \text{ bulk sediment d}^{-1}$. This works out to 8 mmol/L/L/d which is incorrect. This is an issue throughout and needs to be correct. Units on the figures are correct but μmol should be μmol (not plural).

Figure 3: what is the model output based on when there is no geochemical data for a given depth? On line 581, the authors state that only sulfate and methane are considered in the model but it seems to me that the authors assume the behavior of the SMT is the same over time (generally), i.e., the magnitude of AOM is the same over time, it has just migrated up? On line 193, the authors state that methane fluxes change over time but it is not clear how this is implemented. I have a hard time seeing how there is not a time-dependent variability in the methane profile and this should affect the AOM rate. Shouldn't it? Here again the gene data is disconnected to the modeled rate profile. Surely this is telling us something? And the interpretation could be different than what is presented in the paper (i.e. relict biomass)

Line 184: I do not see the 'concave up' shape in these profiles (if so, only between two depths around 10 in 2a and ~35 cm in 3a, certainly not the upper part of the profile in its entirety)

Table S2: It is unclear to me how these numbers were derived? Do these rates reflect modeling of paleo-SMTs? Or present day? Pls clarify.

Line 197 and 222: Since DIC is not 1:1 with ΔSO_4 , AOM may *not* be the only sink for sulfate in some cases. Clarify/explain. Further, sulfate is not completely consumed so what do you mean by “near complete sulfate reduction” (line 224) pathway?

Figure 4: The control core has an AOM rate comparable to the seep cores. How can this be? To me, this suggests an issue with the model? The gene data for this core certainly provides no support for the reported AOM rate.

Line 243: Need to better explain the evidence for how “increases methane flux” and the time scales (years) were determined.

Line 353: I think it would be useful to include cell counts for all of the cores (do these data exist (e.g. ‘high cell densities’ suggest so)

Line 355: seepage is in no way constrained by gas oversaturation; please revise

Line 371: This high flux suggests and incredibly high AOM rate at this site is suspicious...especially given the high sulfate concentration. I suspect this highlights an issue(s) with the model, as the authors suggest. I must admit that this makes me question the model in general, especially in light of the extremely high AOM rates that the model is predicting in general. This issue needs to be resolved because it is very troublesome (I am not a modeler per se, but I have done work on AOM and SMTs and have done limited modeling; something does not seem right here).

Line 404: This is highly speculative and not supported by the data; consider removing.

Line 489: I suggest using great care here because you have modeled AOM rates but you did not ground truth this to assure that the modeled rates compare well with measured rate at reasonable environmental concentrations of electron donor and acceptor. This conclusion needs to be considered in light of various limitations of the data set.

Line 609: I worry that your treatment of GC1048 artificially inflated the methane flux. Can you be sure that was not the case?

Line 628: Can you comment on the extent of contamination (i.e. % of sequences removed as “contamination” and what these sequences most commonly were; would be a help to others doing similar work).

Line 529: Were methane samples collected after splitting cores? If so, surely these values are very much underestimates. Please clarify.

Line 587: This sentence does not make sense.

Figure S3: The regression line is driven by the PC1029 cluster of points to the upper right. This is concerning because the other points in the correlation would drive a much weaker association (if at all).

Figure S4: Here my concerns are presented as a figure – the concave up designation is from the modeled profile, not the empirical data. This must be corrected in the text.

Reviewer #2 (Remarks to the Author):

Review Klasek et al

Methane-driven microbial community succession in Arctic seafloor gas hydrate mounds

In this paper the authors study 7 Arctic seafloor gas hydrate mounds, combining geochemistry with geomicrobiology and reactive transport modelling to discuss the change in the microbial community structure and composition as a function of changes in the methane flux. The key finding of the paper is that at gas seep sites where sulfate concentrations aren't depleted, there are higher populations of anaerobic methanotrophs and sulfate reducing bacteria throughout the sediment column, whereas when the sulfate-methane transition zone is established the population density decreases, and then rebounds when steady state is reached. In theory it is a nice combination of a range of tools used to make an important observation. I suspect within here there is a paper that is a good fit for Nature Communications. But it needs a major reorganization beforehand to make it more understandable to the reader and more logically organized.

Before I get to my criticism of/comments on the paper, it is nearly impossible to give constructive or editorial feedback on a paper that lacks either line numbers or even page numbers. I find myself in a very frustrating position in that I can't direct the authors to where the edits need to go or which parts I wish to comment on. As a result, I will need to use a lot more space than I normally would to write this. Please in the future, for the sake of your reviewers, don't do this again!

Throughout the paper I felt a bit confused as the authors describe the study, for example in the abstract, as a temporal study, I had thought I was reading a paper that looked at sites over different periods of time to watch and monitor microbial populations as a SMT is established (over a multi year study for example). However what the authors have done is to explore sites at the same time and to use the nature of the shape of the sulfate profile to understand how close to 'steady state' the profile is, and, by inference, how long ago the methane was introduced to the sediment. Then this is compared to the geomicrobiological diversity to say something about how community structure and function changes as a result of introduction of methane within a sediment column and the growth of the sulfate-methane transition zone.

The issue with this is that it requires a few leaps of faith, and these aren't explicitly spelled out in the paper until very late, and many in the sections at the end, so it is hard to evaluate them critically. How is the model set up? How are the variables constrained? What is the porosity? Did you include advection or biodiffusion? What are the equations involved? Why model 60 meters of sediment column when the action, as it were, is happening in the top 100-200 cm or shallower? At these shallow depths many other processes with high porosity (not just bioturbation) can become important. I think these assumptions, which are quite central to the premise, need to be brought into the center of the paper to be evaluated as opposed to pinged on the end of the paper. I recognize these were published previously by this group but it would be useful to have them here too in some form!

I suspect that some restructuring will help. On the first page, third paragraph, of the introduction, where you introduce the concept of using the shape of the sulfate profiles to constrain steady state behaviour, consider putting a schematic here as your Figure 1. Or combined with Figure 1 (the map). The schematics come in later but it is hard for me to follow what you are describing here and what you are visualising and since much of the whole concept of the paper lies on this, I think it must come out up front. Before you get to Figures 2, 3, and 4.

In the Results, first paragraph, there is a sentence that starts 6-7 lines down, 'In contrast' – I don't understand this sentence at all. I understand that it is referencing an earlier paper by the group, but 'movement of the aqueous phase at depth' I guess you mean advection? Can you clarify?

Next paragraph first sentence, starting 'Black-coloured' at the end of the sentence you use the word flux but I suspect what you mean is just high rate of production of sulfide. Flux implies the amount that passes a certain interface.

Somewhere in here can you reference that the methods are at the end of the paper?

Again, hard to pin to the paper as I am loath to keep counting the pages and the lines within those pages but I found the last two pages of the results, as a non-specialist, very hard to follow. What is an OTU? What do I take away from figure 6 that is important for the story?

I found Figure 7 to be the most useful figure of the paper and I wonder if it shouldn't come earlier with the change in the microbial diversity results presented in a different way – the schematic that I explained earlier. Not sure what page I am on with Figure 7 but it took me this long to understand the importance and the key findings. It brings it together nicely

Note in your methods you are missing how alkalinity was measured. Can error bars be added to your sulfate concentrations on the plots given the dilution error with the IC?

Finally you get to the main conclusion, that microbial diversity rebounds with the establishment of the SMT, and the suggestion that this is because of the onset of more organoclastic sulfate reduction. I think it must be brought in here that most modelling studies (e.g. Egger et al., Nature Geoscience 2018, Sivan et al. Geobiology 2007) strongly indicate that when there is a steady state SMT that the flux of sulfate down nearly exactly matches the flux of sulfate up and this leaves very little organoclastic sulfate reduction. I didn't see this discrepancy accounted for in the paper?

In short, there is a good paper in here but it needs some reworking to make it a much stronger and much more understandable paper that will have a higher ultimate impact.

Reviewer #3 (Remarks to the Author):

Klasek et al., studied methane fluxes, (anaerobic) methane oxidation and the abundance of AOM performing microorganisms above gas hydrate-bearing sediments of the Storfjordrenna area offshore Svalbard, and they aimed to model the reasons for high AOM using the model of Hong et al. 2017.

The introduction nicely outlines the research subject, from thereon the study becomes quite venturous.

Sampling and Methods:

The methods are insufficient and cannot be followed. Worst are the missing assumptions of the model: The model parameters need to be given, cell size and time steps etc., and complete parameterization, mathematical background etc., what about advection as valid term particular in the vicinity of an actively bubbling seep !

Results:

I read the beginning several times but I cannot see a clear reasoning for the classification of the cores.

For core PC1029 they state that seepage only recently begun (due to the residual sulfate) yet rates should be extremely high (how can organisms establish that quickly). This does not happen in nature for the slow-growing organisms. More likely is that sulfate was carried into the core during sampling - or again advection takes place.

Looking at the modelled rates it is unlikely that the modelling pictures reality. But -see above- it is not possible to validate the model - For instance the authors suggest (over at least 30 cm distance an AOM rate of 4 mmol per liter sediment and day (within in Fig. 2). This would amount to roughly 1.2 M methane consumed per day and square meter - 400 Mol per year - equal to 800 Mol oxic respiration to use up all the produced sulfide.

Looking at the products:

There is not more than 5 mM of sulfide at any depth. Meaning the porewater would need to be exchanged once every 1 day. Also it does not fit with the moderately depleting sulfate concentrations.

Similar sulfate goes down by less than 20 mM. So in lower horizons in half a week the model runs into sulfate depletion - not realistic.

Do we have further indications for such rates -what about advection (to reach that steep gradients necessary for high fluxes) - the vicinity of gas seeps advection is highly likely. How should the amount of sulfate migrate into the sediment? What would happen with all the reaction products?

Also the assumptions in in Model Figure 3 seem to be pretty vague. I am not sure that that the model outcomes are the only solution for the observed geochemistry. Which other parameters have been tested (longer model run) and can you show their outcomes?

For all models I wonder how all reactants and products of AOM behave with time? And are they used to fit the model. I miss additional tested scenarios.

I think without seeing the model it solely relied on diffusion Which scenarios were tested - do you have indications for advective transport and was this tested? I see the term advection only once in the manuscript - in the abstract - what happened?

I cannot evaluate the gene sequence data, yet I am optimistic that the authors can do a SILVA classification of reads.

The discussion of microbial abundance and community composition is arbitrary.

Responses to reviewer comments for manuscript submission NCOMMS-18-28379A

General comments and revisions:

To summarize the changes between the recent version of this manuscript and the one submitted in 2018, we have first considered the reviewers' skepticism of our modeled AOM rates. These summarized three main issues: 1) we did not describe our modeling setup in adequate detail; 2) our interpretations of PC1029 did not consider advection and lead us to calculate very high AOM rates; and 3) prior flux calculations of our steady-state cores did not yield results different from cores where we argued methane flux was increasing. To address these concerns, we have added a supplemental text describing the two modeling scenarios in further detail, remodeled our steady-state cores using a previous approach which obtained lower AOM rates and methane fluxes, and decided not to fit a model to PC1029. These are addressed in the text of the new manuscript.

We have also de-emphasized our interpretation of temporal changes from one methane stage to the next, and decided to refer to them as "regimes" instead. They are presented in the reverse order in the revised manuscript, which we feel reduces confusion for the reader.

We have also reprocessed sequence data differently, in a way that will provide for easier comparison across studies. Instead of clustering sequences as OTUs (operational taxonomic units) at 97% similarity, we obtained exact amplicon sequence variants (ASVs). This did not change our overall conclusions, but we used this opportunity to convey differences in community structure more straightforwardly (Figure 5, showing ASVs whose percent abundances within their communities are different across methane regimes and above/below the SMT, and Figure 7, differences in community structure across methane regimes and redox zones). These replace Figure S2 and Figure 6 in the previous manuscript. These analyses are implemented in reproducible R scripts that are made publicly available on the lead author's github page.

We successively address each reviewer's comments in separate sections below (Bold denotes reviewers' comments).

Reviewer #1

1.1

This paper provides an assessment of methane dynamics in Arctic sediment environments, some of which are impacted to varying degree by methane seepage. Benthic methane fluxes in the Arctic are changing and it is important to understand how microbial communities respond to changes in fluxes. In particular, a key question involves documenting whether anaerobic methane oxidizing communities can keep pace with accelerating methane fluxes. The present paper addresses this important question.

I found this paper to be interesting and well written overall. But I do have some questions regarding a number of the fundamental assumptions and components of the data set. For example, how do the PIs know that no sulfate reduction is fueled by (non-methane) organic matter in the sediments. This is an assumption and it was not validated.

We discount high rates of organoclastic sulfate reduction at these sites based on the low ammonium concentrations. We explain in lines 202-204: ammonium at GC1045 does not exceed 200 μM , and assuming a C:N molar ratio of 6.1 from a previous study in the area, a maximum of only 0.6 mM of sulfate would be consumed by organoclastic sulfate reduction (no ammonium data from GC1081 is available). We have previously demonstrated ammonium profiles in support of a scenario where seafloor methane flux is increasing in Hong et al 2017, *Nature Communications*, *Seepage from an arctic shallow marine gas hydrate reservoir is insensitive to momentary warming*.

This interpretation of AOM as the dominant sink of sulfate in marine sediments, particularly in SMTs experiencing steady-state conditions, is supported by other observations (Egger et al. 2018, *Global diffusive fluxes of methane in marine sediments*, *Nature Geosciences*, and Sivan et al. 2007, *Rates of methanogenesis and methanotrophy in deep-sea sediments*, *Geobiology*).

1.2

Moreover, the authors acknowledge that methane concentrations may be lower than in situ because of degassing during sample collection. However, they ignore the fact that pore fluid profiles could also be altered during degassing. This needs to be mentioned and addressed.

We did not observe any obvious signs of degassing (bubbles or moussy texture) in any of our cores (mentioned in lines 140-141), and only recovered hydrates in one core (PC1029). Still, degassing should only affect porewater profiles below the SMTs, where methane concentrations are expected to be high. The classification of our cores and the modeling depend on an interpretation of the shallow portion of the sulfate profile within the sulfate reduction zone, where no methane degassing is expected due to the overall low methane concentration in the sulfate reduction zone.

1.3

The modeling shows variable methane oxidation rates that are not at all apparent in the pore water profiles and the factors driving these patterns must be assessed.

In general, steep porewater sulfate gradients coincide with high modeled AOM rates, so we assert that these rates are in fact apparent from the profiles. We used a reduced model in the two cores experiencing increased methane flux, which depends on concentrations of methane and sulfate (no other constituents). This setup is described in further detail in our supplemental text (line 43, section titled "Non-steady state model").

For steady-state cores, we have remodeled AOM rates using a previous approach (Hong et al., 2016, *Removal of methane through hydrological, microbial, and geochemical processes in the*

shallow sediments of pockmarks along eastern Vestnesa Ridge (Svalbard), Limnology & Oceanography, described in supplemental information). This approach is also described in our supplemental text (which we reference in lines 646-647), and it results in rates that are nearly an order of magnitude lower than what we presented before. This is evident in Figure S1 (the barplot showing depth-integrated methane fluxes).

Due to large uncertainties in our modeling approach for PC1029 (inability to satisfactorily fit its sulfate profile to an increasing methane flux scenario, or to constrain advection or bioturbation, shown in Figure S7), we have chosen not to present AOM rates. We explain this in lines 238-246, and elaborate further below, in 1.25.

I recommend a major revision for the manuscript and feel the paper *may be* acceptable for publication after major revision. I wish the authors all the best in revising the manuscript; I hope the suggestions below are helpful.

***Specific comments* (suggested insertions/changes are underlined)**

1.4

Line 16: replace 'performing' with mediating
Suggestion accepted, line 20 of new version.

1.5

Line 43: clarify surveying (replace with ...geophysical surveys have characterized...)
Suggestion accepted, line 48 of new version.

1.6

Line 74: This sentence needs clarification. The findings of Kessler et al. are correlative, not causative. It remains unclear whether the oxygen drawdown was due (solely) to methane consumption vs. oil consumption. Notably, Kessler et al. did not quantify methane oxidation rates, they used oxygen as a proxy for methane and that is potentially problematic. Others have suggested that methane was not consumed efficiently during the oil spill.

We have fixed this to convey that methane release, increases in *Gammaproteobacteria*, and oxygen drawdown are correlated and not causative (lines 67-69 of new version).

1.7

Line 79: replace 'is able to' with may
Suggestion accepted, line 72 of new version.

1.8

Line 93: Can you pls add to the legend how far (distance) GC1048 is from the other sites?

We have specified these distances in the legend of Figure 1 (lines 102-103). GC1048 is over 150 m away from the three cores from GHM5 (top panel). This seismic image (bottom panel) shows the position of GC1048 offset from GHM5 and showing no faulting or chimneys, which we interpret to mean fluid flow is very unlikely here.

1.9

Line 100: replace ‘stages in seepage’ with seepage regimes (and define what that means)

We now refer to the three “stages” as “regimes” throughout the manuscript and identify them in the first paragraph of the results in the new submission (lines. 112-129). In particular, we replaced “different stages in seepage” with “where seepage was either observed or absent” (line 113). This helps to minimize our prior portrayal of a temporal succession where one stage leads to another, and replace it with three discrete regimes characterized by whether or not gas bubble emission and advective delivery of seawater into the upper sediment column are present

(lines 127-128), and if not, whether the shape of the sulfate profile is linear or bent (lines 124-125). We also thought it useful to first portray a steady-state biogeochemical equilibrium as the first regime, followed by an increasing methane flux, and then seepage (opposite the way we initially presented the three stages).

1.10

Line 115: Reference core GC1048 is presented as a low flux, non-GHM situation but in figure S1, the present-day methane flux at this site is comparable (higher but no error bars are given) to the other sites. Given this, how can it be a 'control'?

We have since remodeled AOM rates from this core using an existing approach (Hong et al., 2016, *Removal of methane through hydrological, microbial, and geochemical processes in the shallow sediments of pockmarks along eastern Vestnesa Ridge (Svalbard)*, *Limnology & Oceanography*, described in supplemental information in the section titled "Steady State Model...", particularly lines 143-149). Remodeled rates now show that the flux at GC1048 is comparable to those of other steady-state cores (Figure S1). In the original and revised manuscripts, we interpret GC1048 not quite as a control, but as a special case where a steady-state model fits the top 200 cm of the sulfate profile, and the bent shape below 200 cm is fit by a very small increase in methane flux from below (explained further in lines 636-645 of the revised version). We have removed the phrase "reference core GC1048" from the Figure 1 and 2 legends (line 102, 177) to minimize confusion here.

1.11

Figure 1: Sulfide and DIC concentrations are not stoichiometric to what would be expected from AOM and there is more DIC than would be predicted from sulfate drawdown if CH₄ is the sink for sulfate. In fact, sulfate drawdown is about 20 mM but there is over 30 mM DIC at the bottom of the core. It would be useful to compare drawdown of sulfate and production of DIC in relevant horizons to further support the argument that DIC is from AOM.

We assume the reviewer is talking about Figure 2 (which is now Figure 4 in the revised version). We measured alkalinity, not DIC. Though DIC contributes to most of the alkalinity in sediments, smaller fractions come from weak acid-base pairs e.g., sulfide, ammonia, boron, and phosphates. At PC1029, there may be additional alkalinity sourced from deeper methanogenic sediments as a result of silicate weathering as demonstrated in recent studies (e.g., Torres et al. 2020, *Silicate weathering in anoxic marine sediment as a requirement for authigenic carbonate burial*, *Earth Science Reviews*).

1.12

Further, the gene data suggests that AOM related organisms are present throughout the sediment column and show no correlation with the SMT.

The sequence data shows that yes, AOM-related organisms (ANME) are present throughout the sediment columns, but the bubble plots in figures 2C and 3C show that their relative abundances within microbial communities are usually highest at depths near SMTs (or several

cm above). Figure S3 shows a significant positive correlation between *mcrA* gene copy numbers and modeled AOM rates (log-log plot).

1.13

Line 146: I suggest using units for AOM that are more standard and general (i.e. nmol or $\mu\text{mol cm}^{-3} \text{d}^{-1}$ instead of $\mu\text{mol L}^{-1} \text{d}^{-1}$) so that the reader can more easily compare. And, yes, it is a very easy conversion but why make the reader work?

Suggestion accepted, please see figures 2B and 3B, lines 43, 169, and elsewhere in the revised manuscript.

1.14

Line 164: Incorrect units – 8 mM L⁻¹ bulk sediment d⁻¹. This works out to 8 mmol/L/L/d which is incorrect. This is an issue throughout and needs to be correct. Units on the figures are correct but μmol should be μmol (not plural).

Units have been corrected to $\text{nmol cm}^{-3} \text{d}^{-1}$ (line 169 and elsewhere in the revised manuscript), and pluralizations have been corrected likewise.

1.15

Figure 3: what is the model output based on when there is no geochemical data for a given depth? On line 581, the authors state that only sulfate and methane are considered in the model but it seems to me that the authors assume the behavior of the SMT is the same over time (generally), i.e., the magnitude of AOM is the same over time, it has just migrated up? On line 193, the authors state that methane fluxes change over time but it is not clear how this is implemented. I have a hard time seeing how there is not a time dependent variability in the methane profile and this should affect the AOM rate. Shouldn't it?

AOM rates were formulated by multiplying the theoretical maximum rate with the concentrations of methane and sulfate (we have more explicitly reworded this in lines 614-617, and more detail has been added in the supplemental information as Equation 3). This is an expression commonly accepted and applied in similar studies (Regnier et al., 2011, cited within manuscript). With this formulation, AOM rates are a function of methane and sulfate concentrations (while the theoretical maximum rate is kept constant). It is true that the magnitudes of AOM rates do seem to be constant with time, which is due to the high methane supply that constantly maintains AOM at its highest level. However, the slight but noticeable increase in AOM rates at these two cores is attributed to an increase in methane flux from below, which is necessary to fit the observed sulfate profiles.

The underlying assumption of such a setup is that the steepest gradient in the sulfate profile is kept constant with time, which allows us to predict profiles when there is no geochemical data available for the time in the past.

To elaborate more thoroughly about our modeling setup, assumptions, and parameters used, we have added more details in the Supplemental information. Briefly, the model applied to the cores experiencing increasing methane flux (GC1045 and GC1081, section starting at line 44 of the supplemental information) assumes 1) AOM is the only reaction consuming sulfate, justified by low ammonium concentrations in profiles; 2) The non-steady-state porewater profiles are the result of sudden increases in methane flux, as has been put forward in previous research here (Hong et al 2017, cited within manuscript) when ruling out all other non-diffusional processes of fluid movement; and 3) sudden increases in gas supply were due to dissolution and migration through subsurface fractures and phase transitions supported by seismic interpretations.

In addition, we have omitted flux calculations and modeled AOM rate profiles for PC1029, (updated Figure 4) because we cannot constrain bioturbation/bioirrigation by siboglinid tubeworms observed at the site, or bubble-driven advection (explained in lines 233-246).

1.16

(Figure 3): Here again the gene data is disconnected to the modeled rate profile. Surely this is telling us something? And the interpretation could be different than what is presented in the paper (i.e. relict biomass)

We do not necessarily expect the gene data to correlate exactly with modeled rate profiles, which only take into account methane and sulfate concentrations, half-saturation constants for methane and sulfate, and a maximum AOM rate.

We maintain that our *mcrA* gene count data in Figure 3 show peaks at depths very close to the peaks in modeled AOM rate profiles at the time of sampling (black lines), and are not disconnected (lines 294-298, supplemental figure 3). Counts of *dsrAB*, on the other hand, may represent taxa that are not involved in methane cycling if they are not associated with ANME, and thus we would not expect them to be as tightly coupled AOM to rate profiles.

1.17

Line 184: I do not see the ‘concave up’ shape in these profiles (if so, only between two depths around 10 in 2a and ~35 cm in 3a, certainly not the upper part of the profile in its entirety)

The concave-up shape of the sulfate profile can be clearly observed when comparing with Zabel and Schulz 2001, Marine Geology (Importance of submarine landslides for non-steady state conditions in pore water systems - lower Zaire (Congo) deep-sea fan). This is the original paper which systematically studied similar non steady state profiles. (The concave-up description is now found in line 189 of the revised version).

1.18

Table S2: It is unclear to me how these numbers were derived? Do these rates reflect modeling of paleo-SMTs? Or present day? Pls clarify.

We have clarified this in line 192 of the new version: “As estimated by our reduced modeling approach, total methane fluxes throughout these two cores have increased over the past two decades (Table S2).” These were derived from the reduced model now described in the supplemental information.

Briefly, the depth at which we observe a sharp decrease in porewater sulfate allows us to resolve the timing of increased methane flux to approximately a year. Our model output shown here better illustrates this: the following panels indicate a scenario where methane flux has increased 270, 280, 291, or 294 years ago.

1.19

Line 197 and 222: Since DIC is not 1:1 with ΔSO_4 , AOM may *not* be the only sink for sulfate in some cases. Clarify/explain.

As addressed earlier, we measured alkalinity, not DIC. As hinted earlier (1.1), in the recent version (lines 202-204) we justify this assumption because ammonium concentrations from GC1045 are low (not exceeding 200 μM). Assuming a C:N ratio of 10, a maximum of only 1 mM of sulfate would be consumed by organoclastic sulfate reduction. This interpretation of AOM as the dominant sink of sulfate in marine sediments is supported by other observations (Egger et al. 2018, *Global diffusive fluxes of methane in marine sediments*, Nature Geosciences, and Sivan et al. 2007, *Rates of methanogenesis and methanotrophy in deep-sea sediments*, Geobiology). Also, the addition of alkalinity from silicate weathering beneath the SMT will also result in non-1:1 ratios.

1.20

Further, sulfate is not completely consumed so what do you mean by “near complete sulfate reduction” (line 224) pathway?

The “complete sulfate reduction pathway” referred to all biochemical steps involved in dissimilatory sulfate reduction (sulfate oxidation to sulfide). Nevertheless, this sentence was quite speculative, so it has been omitted from the current version.

1.21

Figure 4: The control core has an AOM rate comparable to the seep cores. How can this be? To me, this suggest an issue with the model? The gene data for this core certainly provides no support for the reported AOM rate.

We do not call GC1048 a control core anymore, but after remodeling our rates we now show that the flux at GC1048 is comparable to those of other steady-state cores (Figure S1). We used a two-step approach to model AOM at GC1048, which included a steady-state modeling approach based on eleven primary and eight secondary porewater species (now described in detail in the supplemental text, in the section beginning on line 115). The reduced model (methane and sulfate only) was used to fit the bend in sulfate below 200 cm at GC1048 (lines 145-149 of supplemental text).

Gene abundance data was not used in the model. We found high *mcrA* ($\sim 10^{7.5}$ /g bulk sediment) in GC1048 at 305 cm, which is higher than the other three steady-state cores in figure 2D. The high *mcrA* counts could coincide with moderate to low AOM rates if the methanotrophs are high in abundance, but inactive (discussed in lines 455-461 of the revised version). This sample was from the same depth as the biofilm. Though care was taken not to sample the bulk sediment instead of visible biofilm, cells with high numbers of methanotrophs may be highly distributed among bulk sediments at this depth.

1.22

Line 243: Need to better explain the evidence for how “increases methane flux” and the time scales (years) were determined.

The modeling approach used to justify shoaling of the SMT and increases in methane flux over the past several years are described in the newly-added supplemental text under the subheading “*Non-steady state model for GC1045 and GC1081*”. Also, in the new version of the manuscript, we have specified that the increase in methane flux is over time (line 272). In the revised supplementary information, we have also discussed further how methane fluxes and the time scales, the two unknowns in our model, were determined. Briefly, the flux of methane can be largely determined from the curvature of the sulfate profiles, as higher fluxes will result in a greater sulfate concentration gradient. Once the fluxes are quantified, time scales can be obtained..

1.23

Line 353: I think it would be useful to include cell counts for all of the cores (do these data exist (e.g. 'high cell densities' suggest so)

Unfortunately we did not (and still do not) have cell count data for these samples, and we have stated this in line 403. If *mcrA* concentrations are 10^8 per gram, and taxa that are known to possess *mcrA* (ANME, methanogens, and a few other Archeal phyla) are only a small fraction of the total community, we can indirectly infer high total cell densities.

1.24

Line 355: seepage is in no way constrained by gas oversaturation; please revise

We were trying to convey that seepage could still be present near areas experiencing methane influx (as in GC1081 being very close to seepage on GHM4). Nevertheless, this is a small but distracting detail, so we have omitted this sentence from the current version.

1.25

Line 371: This high flux suggest and incredibly high AOM rate at this site is suspicious...especially given the high sulfate concentration. I suspect this highlights an issue(s) with the model, as the authors suggest. I must admit that this makes me question the model in general, especially in light of the extremely high AOM rates that the model is predicting in general. This issue needs to be resolved because it is very troublesome (I am not a modeler per se, but I have done work on AOM and SMTs and have done limited modeling; something does not seem right here).

We agree our initial calculation of $610 \text{ mol m}^{-1} \text{ yr}^{-1}$ at PC1029 was a surprisingly high rate. In addition to this high rate, during our revision, we have decided that there are too many uncertainties to model AOM rates at this site, including processes of bioturbation and seawater irrigation from vigorous bubbling at the seep site, which may explain the remaining sulfate at depth. Furthermore, we are unable to fit the sulfate porewater profile at PC1029 to a modeling scenario (indicated by Figure S7 in the revised manuscript). Lines 238-244 in the recently submitted manuscript describe our approach:

“As processes other than sulfate diffusion from seawater are not accounted for in our model parameterization, we are unable to precisely calculate AOM rates from PC1029. Our rough estimation of the AOM rate based on the part of the sulfate profile with the greatest concentration gradient (10-15 cmbsf) yields a peak AOM rate on the order of $10^3 \mu\text{moles L}^{-1} \text{ d}^{-1}$. This rate estimate would be increased significantly by accounting for siboglinid-driven pumping of bottom seawater sulfate, or sulfide reoxidation mediated by their endosymbionts”.

This estimated rate from 10-15 cm is now an order of magnitude lower than in our original version.

1.26

Line 404: This is highly speculative and not supported by the data; consider removing.

We believe the lines referenced are: “The high concentrations of *mcrA* seen above the sulfate decline in PC1029 seem unexpected, but AOM rates from 0-5 cm in PC1029 still exceed 80 $\mu\text{mol L}^{-1} \text{d}^{-1}$ (Fig. 2b)”. We have removed the phrasing “seems unexpected” as there is high methane and sulfate present in shallow sediments at PC1029. Perhaps the high rates of AOM are the cause for concern here. This is addressed in the previous comment above. The analogous paragraph of the revised version begins at line 437, discussing the distribution of different clades of ANMEs.

1.27

Line 489: I suggest using great care here because you have modeled AOM rates but you did not ground truth this to assure that the modeled rates compare well with measured rate at reasonable environmental concentrations of electron donor and acceptor. This conclusion needs to be considered in light of various limitations of the data set.

There are only a handful of studies that have compared modeled and empirically measured AOM rates, and it is always tricky to compare the two. For example, Treude et al. 2003, (*Anaerobic oxidation of methane above gas hydrates at Hydrate Ridge, NE Pacific Ocean*, Marine Ecology Progress Series) show that modeled rates are slightly lower than the radiotracer rates. Explanations include that the 1D model may miss lateral methane input, and the radiotracer rates need accurate methane concentration measurements, which are usually lacking. The modeling is fundamentally based on the mass balance of sulfate, so it can be cross-checked with straightforward calculations (Fick’s law for example). The other way to cross-check the modeling results is to compare to other modeling studies (particularly Regnier et al. 2011, who compiled modeling rates and SMT depths), which we have done (lines 409-411 of the revised manuscript).

1.28

Line 609: I worry that your treatment of GC1048 artificially inflated the methane flux. Can you be sure that was not the case?

We have made sure our modeling of GC1048 is much more realistic, given that we have successfully applied this modeling scheme in other locations (please see our answer to 1.10). The methods (lines 636-640) and the supplemental text (lines 143-149) in the revised version provide more detail about the modeling setup of GC1048.

1.29

Line 628: Can you comment on the extent of contamination (i.e. % of sequences removed as “contamination” and what these sequences most commonly were; would be a help to others doing similar work).

We have since identified contaminant sequences in a much more objective and repeatable manner. In the revised version, we used an R package (decontam) that identifies contaminant taxa (amplicon sequence variants, or ASVs) based on whether they were more highly abundant and more prevalent in DNA extraction blanks (this is the “combined” approach in decontam). Overall 81 contaminant ASVs were detected, making up 1.05% of all sequences in the dataset.

These details are specified in lines 670-673. To assist others doing similar work and make it possible for them to identify the exact contaminant sequences, their taxonomy, and their distribution across all samples, we will disseminate the data and code used to identify contaminants on the lead author's github page (Data availability statement on lines 706-710).

1.30

Line 529: Were methane samples collected after splitting cores? If so, surely these values are very much underestimates. Please clarify.

Correct, methane samples were taken after splitting cores. We mentioned splitting cores lengthwise to obtain working and archive halves (line 552) and collecting 5 ml bulk sediment from the working half (lines 556-557).

1.31

Line 587: This sentence does not make sense.

We have fixed this (line 614) to specify the theoretical maximum AOM rate.

1.32

Figure S3: The regression line is driven by the PC1029 cluster of points to the upper right. This is concerning because the other points in the correlation would drive a much weaker association (if at all).

We have removed points from PC1029 in the revised version of this figure, because with our new model, we are less confident about assigning rates to particular depths within this core. We still see a significant positive correlation between *mcrA* copies and AOM rates in Figure S3.

1.33

Figure S4: Here my concerns are presented as a figure – the concave up designation is from the modeled profile, not the empirical data. This must be corrected in the text.

The reader is meant to recognize that the empirical porewater profile and modeled profile (assuming an increasing methane flux) do not match each other for these cores, which justifies their designation as steady-state. To minimize confusion, we have included another supplemental figure (Fig. S7) in the recent manuscript that shows the porewater sulfate profiles overlaid with the best-fit models used to calculate AOM rates. The dashed line at PC1029 shows the poor fit, and justifies our decision to not calculate rates for this core. The red line shows the initial state used as the first-step in modeling GC1048, while the black shows the second step, a slightly increasing methane flux scenario that best fits the modeled profile.

Reviewer #2

In this paper the authors study 7 Arctic seafloor gas hydrate mounds, combining geochemistry with geomicrobiology and reactive transport modelling to discuss the change in the microbial community structure and composition as a function of changes in the methane flux. The key finding of the paper is that at gas seep sites where sulfate concentrations aren't depleted, there are higher populations of anaerobic methanotrophs and sulfate reducing bacteria throughout the sediment column, whereas when the sulfate-methane transition zone is established the population density decreases, and then rebounds when steady state is reached. In theory it is a nice combination of a range of tools used to make an important observation. I suspect within here there is a paper that is a good fit for Nature Communications. But it needs a major reorganization beforehand to make it more understandable to the reader and more logically organized.

2.1

Before I get to my criticism of/comments on the paper, it is nearly impossible to give constructive or editorial feedback on a paper that lacks either line numbers or even page numbers. I find myself in a very frustrating position in that I can't direct the authors to where the edits need to go or which parts I wish to comment on. As a result, I will need to use a lot more space than I normally would to write this. Please in the future, for the sake of your reviewers, don't do this again!

We apologize, and we have made sure to add page and line numbers in the current submission.

2.2

Throughout the paper I felt a bit confused as the authors describe the study, for example in the abstract, as a temporal study, I had thought I was reading a paper that looked at sites over different periods of time to watch and monitor microbial populations as a SMT is established (over a multi year study for example). However what the authors have done is to explore sites at the same time and to use the nature of the shape of the sulfate profile to understand how close to 'steady state' the profile is, and, by inference, how long ago the methane was introduced to the sediment. Then this is compared to the geomicrobiological diversity to say something about how community structure and function changes as a result of introduction of methane within a sediment column and the growth of the sulfate-methane transition zone.

To eliminate some of this confusion, we have since rewritten the manuscript in a way that deemphasizes the temporal succession across the three biogeochemical stages (electing to refer to them as regimes, instead of asserting that one stage necessarily leads to the next). One example: in line 112 of the revised version, "To investigate microbial communities across regimes of methane dynamics" as compared to "To investigate microbial community successional patterns across a range of methane seepage activity" in the previous version. Importantly, we acknowledge the high ANME/SRB populations at the seep site are likely a result of consistent methane supply over a long period of time (years), rather than a recent influx of methane into this zone (lines 421 of the revised version).

However, within the regime of increasing methane flux, we still use reactive-transport modeling of AOM rates to infer the upward diffusion of methane into shallower depths, and the microbial

community response to this. We depict this in the figure below, where the following panels indicate a scenario where methane flux has increased 270, 280, 291, or 294 years ago.

In addition, we have reordered the regimes to make more sense with what one could reasonably expect to happen in succession (steady-state, then increasing methane flux, then seepage, mentioned in lines 120-128) though we do not outright state this.

2.3

The issue with this is that it requires a few leaps of faith, and these aren't explicitly spelled out in the paper until very late, and many in the sections at the end, so it is hard to evaluate them critically. How is the model set up? How are the variables constrained? What is the porosity? Did you include advection or bioturbation? What are the equations involved? Why model 60 meters of sediment column when the action, as it were, is happening in the top 100-200 cm or shallower? At these shallow depths many other processes with high porosity (not just bioturbation) can become important. I think these assumptions, which are quite central to the premise, need to be brought into the center of the paper to be evaluated as opposed to pinged on the end of the paper. I recognize these were published previously by this group but it would be useful to have them here too in some form!

To address these concerns, we have added text to our supplemental information (which we reference in lines 645-647 of the main text) that explains these models in more detail (including explicit assumptions at lines 47-72, porosity of 0.7 at line 77, and equations at lines 75-76 and

101). The model considers diffusion, and does not include any advection, irrigation, or any diffusion brought on by bioturbation. The 60 m sediment column was modeled because the 60 m depth corresponds to the base of the gas hydrate stability zone at Storfjordrenna (lines 45-46).

Briefly, there are two reactive-transport models for the different regimes modeled: a steady-state model that factors in 18 porewater constituents, and a reduced model that considers methane and sulfate only, assuming methane as the only sink for sulfate. Best-fit models of sulfate profiles for all cores are shown in Figure S7. Two cores presented special cases. Due to its slight curvature below 200 cm, the full model was used on the top portion of GC1048, while the curve was fitted with a slightly increasing methane flux scenario from the reduced model (lines 145-149 of supplemental text). PC1029 could not be satisfactorily fit with either modeling approach, and for this reason we have decided not to present AOM rates (see updated Figure 4). We provide an estimate for the depth interval corresponding to the steepest decline in the sulfate profile (lines 240-242 in the revised manuscript), and more critically, a reinterpretation of fluid flow at this core that considers bioirrigation or bubble-driven advection (lines 229-240).

2.4

I suspect that some restructuring will help. On the first page, third paragraph, of the introduction, where you introduce the concept of using the shape of the sulfate profiles to constrain steady state behaviour, consider putting a schematic here as your Figure 1. Or combined with Figure 1 (the map). The schematics come in later but it is hard for me to follow what you are describing here and what you are visualising and since much of the whole concept of the paper lies on this, I think it must come out up front. Before you get to Figures 2, 3, and 4.

We think this is a helpful suggestion and have added schematics to supplement the map in Figure 1 (panels 1B-D).

2.5

In the Results, first paragraph, there is a sentence that starts 6-7 lines down, ‘In contrast’ – I don’t understand this sentence at all. I understand that it is referencing an earlier paper by the group, but ‘movement of the aqueous phase at depth’ I guess you mean advection? Can you clarify?

We have changed the wording of this sentence to: “In contrast, abundant trace elements in porewaters from GHM5 point towards upward fluid advection, which may have followed prior seepage activity” (lines 116-118 of revised manuscript). This was in the past, as we do not mean to convey that there was advective fluid flow at GHM5 at the time of sampling (in fact, cores from GHM5 are dominated by diffusion).

2.6

Next paragraph first sentence, starting ‘Black-coloured’ at the end of the sentence you use the word flux but I suspect what you mean is just high rate of production of sulfide. Flux implies the amount that passes a certain interface.

Good point, we have changed the wording to: “Black-colored glaciomarine sediments were recovered in all cores, reflecting the precipitation of iron sulfide minerals resulting from high rates of sulfide production” (lines 131-132 of the revised manuscript).

2.7

Somewhere in here can you reference that the methods are at the end of the paper?

Suggestion accepted, we have mentioned this in lines 157-158 of the revised manuscript.

2.8

Again, hard to pin to the paper as I am loath to keep counting the pages and the lines within those pages but I found the last two pages of the results, as a non-specialist, very hard to follow. What is an OTU? What do I take away from figure 6 that is important for the story?

The original Figure 6 attempted to show differences across communities across redox zones and stages of methane dynamics. We have eliminated it and replaced it with Figures 5 and 7 in the new manuscript.

OTU stands for “Operational Taxonomic Unit”, which is a cluster of sequences that are typically more than 97% similar to one another, essentially a species-level taxonomic assignment. They’re obsolete now, as ASVs (Amplicon Sequence Variants, line 144) are exact sequences that are not clustered and can thus be compared across datasets. We reprocessed our sequence data as ASVs and found that community patterns were conserved (i.e. alpha-diversity trends in the new Figure 6 are essentially unchanged), though we are now able to convey all communities as an ordination in 2D space (Figure 7). This shows much more unambiguously that microbial community structure varies across methane regimes and geochemical zones (explained further in lines 339-355). Within this paragraph we discuss the follow-up statistical tests across groups of samples (PERMANOVAs) that reinforce the clustering patterns seen in Figure 7.

Figure 6 of the old manuscript also showed taxonomy of OTUs with higher relative abundances in certain regimes or redox zones. This information has been replaced by Figure 5, which also conveys how strongly each ASV associates with a particular regime (above or below the SMT) and its average percent abundance within the regime/zone it associates with. This is explained in lines 306-315.

I found Figure 7 to be the most useful figure of the paper and I wonder if it shouldn’t come earlier with the change in the microbial diversity results presented in a different way – the schematic that I explained earlier. Not sure what page I am on with Figure 7 but it took me this long to understand the importance and the key findings. It brings it together nicely

Thank you! Note that the conceptual figure is now Figure 8 in the revised manuscript, and again, we do not specify a succession from one stage to the next.

2.9

Note in your methods you are missing how alkalinity was measured.

Alkalinity was measured shipboard, so we included it under the subheading “fieldwork and sample collection” in lines 558-564 of the revised version.

2.10

Can error bars be added to your sulfate concentrations on the plots given the dilution error with the IC?

The error should actually be around 3-4% for our sulfate measurements (updated in lines 586-588). As such, we have added +/- 4% error bars on sulfate measurements, though they are barely visible beyond the sizes of the points. They are visible in Figure 4, and so they are mentioned in the figure legend (line 265).

2.11

Finally you get to the main conclusion, that microbial diversity rebounds with the establishment of the SMT, and the suggestion that this is because of the onset of more organoclastic sulfate reduction. I think it must be brought in here that most modelling studies (e.g. Egger et al., Nature Geoscience 2018, Sivan et al. Geobiology 2007) strongly indicate that when there is a steady state SMT that the flux of sulfate down nearly exactly matches the flux of sulfate up and this leaves very little organoclastic sulfate reduction. I didn't see this discrepancy accounted for in the paper?

We believe this comment refers to our sentence "Organic matter fermentation, mediated by Atribacteria, may be more dominant in steady-state sites, while Sulfurovum, which oxidizes elemental sulfur or thiosulfate using oxygen or nitrate as electron acceptor, may mediate this cycle above SMTs in non-steady-state areas". We do not claim that the rebound of microbial diversity or the increase in abundance of certain taxa is associated with organoclastic sulfate reduction. In fact, we mention that Atribacteria are known to dominate communities below SMTs, and genomic evidence suggests they are likely fermenters (anaerobic heterotrophs using organic compounds as electron acceptors).

Nevertheless, after reprocessing our sequence data using amplicon sequence variants (ASVs) instead of operational taxonomic units (OTUs), we see these taxa are more evenly distributed across regimes than we previously thought, so we eliminated the sentence. We still see the same reduction in within-community diversity across peak AOM depths (lines 325-328). Instead of implying successional changes in microbial communities from one regime to the next, we present the few taxa (ASVs) that are differentially abundant across methane regimes, with a few potential explanations (lines 490-510 of revised version).

In short, there is a good paper in here but it needs some reworking to make it a much stronger and much more understandable paper that will have a higher ultimate impact.

Reviewer #3

Klasek et al., studied methane fluxes, (anaerobic) methane oxidation and the abundance of AOM performing microorganisms above gas hydrate-bearing sediments of the Storfjordrenna area offshore Svalbard, and they aimed to model the reasons for high AOM using the model of Hong et al. 2017.

The introduction nicely outlines the research subject, from thereon the study becomes quite venturous.

3.1

Sampling and Methods:

The methods are insufficient and cannot be followed. Worst are the missing assumptions of the model: The model parameters need to be given, cell size and time steps etc., and complete parameterization, mathematical background etc., what about advection as valid term particular in the vicinity of an actively bubbling seep !

To address these concerns, we have added text to our supplemental information (which we reference in lines 646-647) that explains the model setup in more detail (including porosity, equations used, and explicit assumptions, lines 47-85 in the supplemental text). The model considers diffusion, and does not include any advection, irrigation, or any diffusion brought on by bioturbation. PC1029 could not be satisfactorily fit with either modeling approach, and for this reason we have decided not to present AOM rates (see the revised Figure 4). We provide an estimate for the depth interval corresponding to the steepest decline in the sulfate profile (lines 240-242 in the revised manuscript), and more critically, we provide a reinterpretation of fluid flow at this site that considers bioirrigation or bubble-driven advection (lines 233-240).

Briefly, there are two reactive-transport models for the different regimes modeled: a steady-state model that factors in nineteen porewater constituents, and a reduced model that considers methane and sulfate only, assuming methane as the only sink for sulfate (each are detailed in respective sections in the supplemental text). Best-fit models of sulfate profiles for all cores are shown in Figure S7. Two cores presented special cases. Due to its slight curvature below 200 cm, the full model was used on the top portion of GC1048, while the curve below was fitted with a slightly increasing methane flux scenario from the reduced model (lines 636-644 of the revised text, and lines 145-149 of the supplemental text). PC1029 was not modeled.

Time discretization of 0.02 years and cell sizes of 1 cm were used for the full model, as specified in the supplemental information of Hong et al 2016, *Removal of methane through hydrological, microbial, and geochemical processes in the shallow sediments of pockmarks along eastern Vestnesa Ridge (Svalbard)*, Limnology & Oceanography. The reduced model used a time discretization of 0.01 year and a cell depth of 0.025 m, as specified in the methods of Hong et al 2017 Nature Communications, *Seepage from an arctic shallow marine gas hydrate reservoir is insensitive to momentary ocean warming*.

3.2

Results:

I read the beginning several times but I cannot see a clear reasoning for the classification of the cores.

For core PC1029 they state that seepage only recently begun (due to the residual sulfate) yet rates should be extremely high (how can organisms establish that quickly). This does not happen in nature for the slow-growing organisms. More likely is that sulfate was carried into the core during sampling - or again advection takes place.

We have rewritten the manuscript in a way that deemphasizes the temporal succession across the three biogeochemical stages (electing to refer to them as regimes, instead of asserting that one stage necessarily leads to the next). Regimes are grouped according to biogeochemical patterns described in lines 123-128 of the revised manuscript. These include the linearity of

porewater sulfate profiles, magnitudes of AOM rates and fluxes, patterns in microbial community diversity and composition, and abundances of key genes involved in SR/AOM.

Importantly, we reinterpret the high ANME/SRB populations at the seep site (PC1029) as a likely result of consistent methane supply over a long period of time, rather than a recent influx of methane into this zone (lines 419-421 of the revised version). We reinterpret the high sulfate concentrations above 10 cm in PC1029 as a consequence of bioturbation or bubble-driven convection (lines 233-236 of the revised version). We thank you for the helpful comments and interpretations.

3.3

Looking at the modelled rates it is unlikely that the modelling pictures reality. But -see above- it is not possible to validate the model - For instance the authors suggest (over at least 30 cm distance an AOM rate of 4 mmol per liter sediment and day (within in Fig. 2). This would amount to roughly 1.2 M methane consumed per day and square meter - 400 Mol per year – equal to 800 Mol oxic respiration to use up all the produced sulfide.

Looking at the products:

There is not more than 5 mM of sulfide at any depth. Meaning the porewater would need to be exchanged once every 1 day. Also it does not fit with the moderately depleting sulfate concentrations.

Similar sulfate goes down by less than 20 mM. So in lower horizons in half a week the model runs into sulfate depletion - not realistic.

The high rates we originally presented at PC1029 puzzled us, and in trying to reinterpret and remodel this core we have come to the conclusion that advection of seawater is likely occurring, and we cannot adequately fit a scenario to the sulfate profile (shown by Figure S7 in the revised supplementals). We address this setup and its uncertainties in the discussion (lines 411-421) of the revised manuscript.

3.4

Do we have further indications for such rates –what about advection (to reach that steep gradients necessary for high fluxes) - the vicinity of gas seeps advection is highly likely. How should the amount of sulfate migrate into the sediment? What would happen with all the reaction products?

Aside from PC1029, we negate advection because our previous work on cores from Storfjordrenna GHMs indicates that advective scenarios do not concur with measured profiles of sulfate, sulfide, alkalinity, calcium, ammonium, and iron (II), in contrast to scenarios describing increasing methane flux. This is illustrated in figure 4 from Hong et al, 2017 Nature Communications, *Seepage from an arctic shallow marine gas hydrate reservoir is insensitive to momentary ocean warming*. Thus we feel confident negating advection in the modeling presented here.

3.5

Also the assumptions in in Model Figure 3 seem to be pretty vague. I am not sure that that the model outcomes are the only solution for the observed geochemistry. Which other parameters have been tested (longer model run) and can you show their outcomes? For all models I wonder how all reactants and products of AOM behave with time? And are they used to fit the model. I miss additional tested scenarios. I think without seeing the model it solely relied on diffusion Which scenarios were tested – do you have indications for advective transport and was this tested? I see the term advection only once in the manuscript – in the abstract – what happened?

We have added a more detailed description of our reduced model and its assumptions in the supplemental text (section starting at line 44). Again, other modeling scenarios were tested and discounted in Hong et al 2017. As mentioned in our previous comments, we now consider advection is contributing to high sulfate concentrations in PC1029.

3.6

I cannot evaluate the gene sequence data, yet I am optimistic that the authors can do a SILVA classification of reads.

The discussion of microbial abundance and community composition is arbitrary.

16S microbial community sequencing is a routine procedure, and we have reprocessed our sequence data and followed state-of-the-art methods for sequencing, sequence processing, analyzing alpha- and beta-diversity within and across communities, and identifying differentially-abundant sequences between groups of samples.

REVIEWER COMMENTS

Reviewer #2 (Remarks to the Author):

This is the second time I have seen this paper and it is improved on my previous reading of it (although that was a few years ago now. I thank the authors for taking my comments on board and in particular for including line numbers.

I have included minor comments below, but the two major ones are that for a study that relies on reactive transport modelling in non-steady state conditions, the fact that the sampling frequency for pore fluid concentrations is so very very low, means I lose confidence in the conclusions. In some sites there is a sample every meter below the sediment-water interface. I know that, now that the samples are acquired, you can't necessarily go back and make a high resolution profile of changes in sulfate concentration, with analyses every five cm to be able to properly figure out the degree of linearity or concavity as speculated, but I do think therefore this means that this should be acknowledged as a limitation, or the reactive transport modelling made a less significant part of the paper, or the reactive transport modelling should be done on three hypothetical scenarios rather than on the actual data, since the actual data is incomplete.

Of these I like the last the best, personally, as I think the modelling oversells what is actually there right now. Modelling based on sulfate and methane concentration profiles as hypothesised for these cores would give better confidence to what I consider the most important contribution of this work, the change in the rates of AOM calculated or suggested.

I also don't like the use of the word 'regime' which I think is vague. I would consider giving names to the three states - "steady state", "enhanced methane flux" and "gas seep" rather than describing as regimes. Again this speaks to the lack of high resolution data. In reality these exist on a continuum - not as 'regimes' persay. It is just how you have grouped them.

Line 22 - the flux of methane rather than methane flux.

Line 25 - remove 'dynamics'

Line 42-43 I am not sure these rates are particularly useful in the absence of other information. Can you put these rates in context with other rates or remove this fact? Or state how much of the methane oxidation is anaerobic versus aerobic?

Line 46 - I would stick with hydrosphere rather than atmosphere as it isn't clear that all of this methane makes it to the atmosphere and isn't consumed in the water column.

Line 60 - forcing should be singular.

Lines 66-80 - The paper in PNAS by Contreras et al should be considered and cited here somewhere.

Line 78 - sulfate concentrations decrease (not sulfate decreases)

Line 84-85 - Suggest you also cite Turchyn et al, Global Planetary Change, 2016 as an example of a methane and sulfate profile not in steady state.

Line 93 - 'rates of AOM' rather than 'AOM rates'

Line 94 - ditto - 'abundances of ANME/SRB' rather than ANME/SRB abundances

Line 112 - I am not sure that 'regimes of methane dynamics' is very clear here. "across various deviations from steady state?"

Line 116 - what is meant by limited movement in porewater? Presumably the gas is moving in the pore water?

Line 119, 310, 364 - again regimes doesn't feel like the right word.

Line 120-129 - consider moving this to the top of this section. It feels like the introduction to the results is here.

Line 137 - decline in sulfate concentration

Line 139 - concentrations rather than values

Line 140 - what is moussy sediment texture?

Line 141 - Would be a bit more cautious here. Suggesting limited degassing on core retrieval. Not

sure the possibility is discounted entirely.

Line 162-165 – didn't you say this earlier in the results section?

Line 165 – I'd use 'track' rather than 'mirror' as 'mirror' can be used to mean the opposite whereas what you mean is that they do the same thing.

Line 168 – rates of AOM rather than AOM rates.

Line 189 – suggesting rather than indicating

Line 192 – what is reduced modelling?

Line 202-204 – So this assumes hydrogenotrophic methanogenesis rather than acetoclastic? I can't quite make the link you are making with the ammonia concentrations.

Figure 3 – how do you know that what you are looking at is not advection? These look like advection profiles at the top to me.

Line 328 – add 'the' before 'highest'

Line 393 – remove 'dynamics'

Line 394 – add 'the' before 'methane'

Line 393 – The way this is written suggests it never was at steady state. My take on this is that it was at steady state and then perturbed through an increased rate of methane advection from below? Can you rephrase with this in mind.

Line 411- within the middle of what? Clarify.

Reviewer #4 (Remarks to the Author):

In this study, Klasek et al. investigated the microbial community composition in a limited number of sediment cores (pushcores and gravity cores) using 16S rRNA gene sequencing and digital droplet PCR targeting *dsrAB* and *mcrA* genes. They associated the resulting depth distributions of potential anaerobic methane oxidizers and sulfate reducers to geochemical profiles and numerical models to propose different patterns of microbial response to changes in regime of methane fluxes.

Although the revisions of the manuscript slightly improved the overall quality of the paper I am still doubtful regarding the validity of the model and the interpretation of the results. The model is tested on a limited number of cores and required a faith that is not compatible with the high standard of Nature Communications. Assuming that sulfate reduction is only associated with methane oxidation is an aberration when AOM-independent sulfate reducing lineages are detected (e.g.: *desulfatiglans*, *Chloroflexi* ...). FISH microscopy experiments witnessing that all/or at least most of the SRB were associated to ANME are in my opinion mandatory. Genomic data might also be useful to evaluate the full diversity of SRB in the system. The interpretation of the results is based on quantification of genes and therefore does not reflect the in situ activity. Lag between DNA and RNA results are frequently observed in cold seeps; this should be taken into consideration. Authors assumed that only cell division and growth can explain an increase of relative abundance but microbial aggregates can be transported in porewater over centimeters, particularly in high fluid flow.

Reviewer Comments

Reviewer #2 (Remarks to the Author):

This is the second time I have seen this paper and it is improved on my previous reading of it (although that was a few years ago now). I thank the authors for taking my comments on board and in particular for including line numbers.

I have included minor comments below, but the two major ones are that for a study that relies on reactive transport modelling in non-steady state conditions, the fact that the sampling frequency for pore fluid concentrations is so very very low, means I lose confidence in the conclusions. In some sites there is a sample every meter below the sediment-water interface. I know that, now that the samples are acquired, you can't necessarily go back and make a high resolution profile of changes in sulfate concentration, with analyses every five cm to be able to properly figure out the degree of linearity or concavity as speculated, but I do think therefore this means that this should be acknowledged as a limitation, or the reactive transport modelling made a less significant part of the paper, or the reactive transport modelling should be done on three hypothetical scenarios rather than on the actual data, since the actual data is incomplete.

Of these I like the last the best, personally, as I think the modelling oversells what is actually there right now. Modelling based on sulfate and methane concentration profiles as hypothesized for these cores would give better confidence to what I consider the most important contribution of this work, the change in the rates of AOM calculated or suggested.

I also don't like the use of the word 'regime' which I think is vague. I would consider giving names to the three states – "steady state", "enhanced methane flux" and "gas seep" rather than describing as regimes. Again, this speaks to the lack of high resolution data. In reality these exist on a continuum – not as 'regimes' persay. It is just how you have grouped them.

Line 22 – the flux of methane rather than methane flux.

Line 25 – remove 'dynamics'

Line 42-43 I am not sure these rates are particularly useful in the absence of other information. Can you put these rates in context with other rates or remove this fact? Or state how much of the methane oxidation is anaerobic versus aerobic?

Line 46 – I would stick with hydrosphere rather than atmosphere as it isn't clear that all of this methane makes it to the atmosphere and isn't consumed in the water column.

Line 60 – forcing should be singular.

Lines 66-80 – The paper in PNAS by Contreras et al should be considered and cited here somewhere.

Line 78 – sulfate concentrations decrease (not sulfate decreases)

Line 84-85 – Suggest you also cite Turchyn et al, Global Planetary Change, 2016 as an example of a methane and sulfate profile not in steady state.

Line 93 – ‘rates of AOM’ rather than ‘AOM rates’

Line 94 – ditto – ‘abundances of ANME/SRB’ rather than ANME/SRB abundances

Line 112 – I am not sure that ‘regimes of methane dynamics’ is very clear here. “across various deviations from steady state?”

Line 116 – what is meant by limited movement in porewater? Presumably the gas is moving in the pore water?

Line 119, 310, 364 – again regimes doesn’t feel like the right word.

Line 120-129 – consider moving this to the top of this section. It feels like the introduction to the results is here.

Line 137 – decline in sulfate concentration

Line 139 – concentrations rather than values

Line 140 – what is moussy sediment texture?

Line 141 – Would be a bit more cautious here. Suggesting limited degassing on core retrieval. Not sure the possibility is discounted entirely.

Line 162-165 – didn’t you say this earlier in the results section?

Line 165 – I’d use ‘track’ rather than ‘mirror’ as ‘mirror’ can be used to mean the opposite whereas what you mean is that they do the same thing.

Line 168 – rates of AOM rather than AOM rates.

Line 189 – suggesting rather than indicating

Line 192 – what is reduced modelling?

Line 202-204 – So this assumes hydrogenotrophic methanogenesis rather than acetoclastic? I can’t quite make the link you are making with the ammonia concentrations.

Figure 3 – how do you know that what you are looking at is not advection? These look like advection profiles at the top to me.

Line 328 – add ‘the’ before ‘highest’

Line 393 – remove ‘dynamics’

Line 394 – add ‘the’ before ‘methane’

Line 393 – The way this is written suggests it never was at steady state. My take on this is that it was at steady state and then perturbed through an increased rate of methane advection from below? Can you rephrase with this in mind.

Line 411- within the middle of what? Clarify.

Reviewer 2 response

We thank you for your helpful comments. Given the low depth resolution of our porewater data, we have decided to take your suggestion to model AOM rates from hypothetical profiles of sulfate and methane. This approach is what we presented in the previous version of the manuscript as the “reduced model” for non-steady-state cores; this is the only modeling we present in the updated version. As we cannot rule out advection in our seep site (PC1029), we maintain that there is too much uncertainty to estimate an AOM rate profile. But in non-steady and steady-state cores, we see large differences in model run times that produce sulfate profiles that correspond closely with our data. (10^2 years for non-steady-state cores, compared to $\sim 10^4$

yrs for steady-state, Figure 2 and highlighted text in lines 144-169 in the updated manuscript). This gives us additional confidence in our classification scheme that we elaborate on in the rest of the manuscript. Please note that we are now considering GC1048 as a third “transitional” state due to the slight bend in its porewater sulfate profile below 2.5 m (Fig. 2K, lines 159-164). This fits a different modeling scenario of an increasing methane flux, though with much lower magnitude ($<1 \text{ mol m}^2 \text{ yr}^{-1}$ compared to 4-5) and lower SMT shoaling rate ($0.4 \text{ cm}^{-1} \text{ yr}$ compared to 10). As such, we have placed porewater, community, and gene abundance data from this core separately in supplemental figure S3, and omitted communities from this core in subsequent analyses which depend on steady-state vs non-steady-state classifications.

In addition to adding a figure showing differences in model output between steady-state, transitional, and non-steady-state cores (Figure 2), we have removed AOM rate profiles from steady-state cores (Figure 3) because they were fitted to our sparse porewater data. We have also removed descriptions and figures showing the fitted modeling approach from the main and supplemental text.

We have agreed to refer to the different patterns of methane dynamics as “states” instead of “regimes” throughout the text of the manuscript and supplementals. (We had previously used “stages”, but backed away from that term because of the unintended implication that one stage should deterministically lead to the next). However, we prefer the term “non-steady-state” instead of “enhanced methane flux” because we feel it better conveys the imbalance between methane and sulfate fluxes that result in a shoaling of the SMT with time, which is a pattern we also emphasize in our microbial data. We sometimes redundantly refer to these cores as “non-steady-state cores experiencing increased methane flux” for emphasis.

Inline responses:

Line 22 – the flux of methane rather than methane flux.

Suggestion accepted, now line 23.

Line 25 – remove ‘dynamics’

Suggestion accepted

Line 42-43 I am not sure these rates are particularly useful in the absence of other information. Can you put these rates in context with other rates or remove this fact? Or state how much of the methane oxidation is anaerobic versus aerobic?

We have omitted the rates here, as they are put into context later in the manuscript (particularly the paragraph beginning at line 423). We also clarified that the consumption refers to both anaerobic and aerobic methane oxidation, line 43).

Line 46 – I would stick with hydrosphere rather than atmosphere as it isn’t clear that all of this methane makes it to the atmosphere and isn’t consumed in the water column.

Suggestion accepted, now line 47.

Line 60 – forcing should be singular.

Suggestion accepted, now line 62.

Lines 66-80 – The paper in PNAS by Contreras et al should be considered and cited here somewhere.

Though interesting, we do not feel the Contreras PNAS manuscript should be cited in this paragraph, as these studies discuss much shorter-term microbial community dynamics under changing methane conditions as supported by molecular sequencing methods.

Line 78 – sulfate concentrations decrease (not sulfate decreases)

We have generalized this to “infer rates and fluxes of sulfur through sulfate-reducing bacterial communities” in lines 80-81 in accordance with the Turchyn study.

Line 84-85 – Suggest you also cite Turchyn et al, Global Planetary Change, 2016 as an example of a methane and sulfate profile not in steady state.

We have included this study as another example of how modeling of sulfate profiles can reveal information about microbial activities, line 81.

Line 93 – ‘rates of AOM’ rather than ‘AOM rates’

Suggestion accepted, now line 101.

Line 94 – ditto – ‘abundances of ANME/SRB’ rather than ANME/SRB abundances

Suggestion accepted, now line 101.

Line 112 – I am not sure that ‘regimes of methane dynamics’ is very clear here. “across various deviations from steady state?”

We have rephrased these paragraphs to better convey how the modeling informs our classification of these states of methane dynamics. Descriptions of steady and non-steady states are provided in paragraphs beginning on lines 155 and 166, respectively (highlighted text).

Line 116 – what is meant by limited movement in porewater? Presumably the gas is moving in the pore water?

We meant to convey that most of the methane at seep sites is transported through the gaseous phase, as opposed to aqueous. We have clarified that this advective transport (whether through bubble movement or bioturbation) prevents us from using our diffusion-based model on the seep core, justifying it as a separate state (lines 177-179, highlighted).

Line 119, 310, 364 – again regimes doesn’t feel like the right word.

We have replaced mention of “regimes” with “states” throughout the text, and in particular, carefully restructured how we define these stages in these paragraphs (lines 143-179).

Line 120-129 – consider moving this to the top of this section. It feels like the introduction to the results is here.

In revising this manuscript, we have decided to first introduce how our modeling setup defines different states of methane dynamics (lines 144-153), and then dedicate separate paragraphs to each state (lines 155-179). We also felt it made more sense to present “field results and general patterns” before “identification of distinct states”.

Line 137 – decline in sulfate concentration

Suggestion accepted, now lines 121-122.

Line 139 – concentrations rather than values

Suggestion accepted, now line 123.

Line 140 – what is moussy sediment texture?

Resembling the texture of a chocolate mousse. Nevertheless, changed moussy to “frothy”, now line 124.

Line 141 – Would be a bit more cautious here. Suggesting limited degassing on core retrieval. Not sure the possibility is discounted entirely.

Changed “discounting” to “limiting”, now line 125.

Line 162-165 – didn’t you say this earlier in the results section?

Correct, we omitted this redundant information referring to our classification of steady-state cores (now in lines 155-158).

Line 165 – I’d use ‘track’ rather than ‘mirror’ as ‘mirror’ can be used to mean the opposite whereas what you mean is that they do the same thing.

Suggestion accepted, now line 198.

Line 168 – rates of AOM rather than AOM rates.

Suggestion accepted, note that these AOM rates and methane fluxes now derive from the modeling approach without fitting sparse sulfate data (lines 201-202).

Line 189 – suggesting rather than indicating

Suggestion accepted, now line 220.

Line 192 – what is reduced modelling?

“Reduced modeling” does not attempt to fit profiles to sulfate porewater data, in contrast to the full modeling approach used to fit steady-state cores in the previous version of the manuscript. We have removed this clarification since deciding to apply the model to hypothetical scenarios, as you have suggested, so now all our modeling results shown are from the “reduced model”.

Line 202-204 – So this assumes hydrogenotrophic methanogenesis rather than acetoclastic? I can’t quite make the link you are making with the ammonia concentrations.

We are not discussing methanogenesis here—we brought up the ammonium to make the point that AOM is the dominant sink of sulfate. If sulfate were being reduced with organic matter, we would expect a much higher buildup of ammonium concentrations downcore. We now acknowledge this assumption earlier in the text as we describe different states (lines 147-149) and justify AOM as the dominant sulfate sink in the supplemental information, particularly Figure S2 and Table S3). On a separate note, the slight increase in ammonium observed in non-steady-state cores allows us to discount the advective intrusion of oxic bottom water (lines 229-231).

Figure 3 – how do you know that what you are looking at is not advection? These look like advection profiles at the top to me.

In addition to observing a slight ammonium increase downcore in GC1045, previous modeling work (Hong et al 2017, Nature Communications) on some of the non-steady-state cores at Storfjordrenna examined the possibility of advection, but noted that an advective scenario failed to fit the observed sulfate, magnesium, and calcium profiles. We have clarified this in lines 225-228, and added a supplementary table that shows depths and concentrations of maximum porewater ammonium from all cores collected from the Storfjordrenna area (Table S3).

Line 328 – add ‘the’ before ‘highest’

Suggestion accepted, now line 344.

Line 393 – remove ‘dynamics’

We have omitted this, and summarized non-steady-state characteristics more specifically here (lines 410-412).

Line 394 – add ‘the’ before ‘methane’

We have since reworded this part.

Line 393 – The way this is written suggests it never was at steady state. My take on this is that it was at steady state and then perturbed through an increased rate of methane advection from below? Can you rephrase with this in mind.

Thanks for pointing this out, this was a holdover from a previous draft where we presented the seep first, then non-steady-state. The model does indeed assume a steady-state as an initial condition (Fig. S1A). We have rephrased our non-steady-state as a departure from a steady-state condition in lines 405-414).

Line 411- within the middle of what? Clarify.

Rephrased to “average flux values”, instead of “middle”, now line 429.

Reviewer #4 (Remarks to the Author):

In this study, Klasek et al. investigated the microbial community composition in a limited number of sediment cores (pushcores and gravity cores) using 16S rRNA gene sequencing and digital droplet PCR targeting *dsrAB* and *mcrA* genes. They associated the resulting depth distributions of potential anaerobic methane oxidizers and sulfate reducers to geochemical profiles and numerical models to propose different patterns of microbial response to changes in regime of methane fluxes.

Although the revisions of the manuscript slightly improved the overall quality of the paper I am still doubtful regarding the validity of the model and the interpretation of the results. The model is tested on a limited number of cores and required a faith that is not compatible with the high standard of Nature Communications. Assuming that sulfate reduction is only associated with methane oxidation is an aberration when AOM-independent sulfate reducing lineages are detected (e.g.: *Desulfatiglans*, *Chloroflexi* ...). FISH microscopy experiments witnessing that all/or at least most of the SRB were associated to ANME are in my opinion mandatory. Genomic data might also be useful to evaluate the full diversity of SRB in the system. The interpretation of the results is based on quantification of genes and therefore does not reflect the in situ activity. Lag between DNA and RNA results are frequently observed in cold seeps; this should be taken into consideration. Authors assumed that only cell division and growth can explain an increase of relative abundance but microbial aggregates can be transported in porewater over centimeters, particularly in high fluid flow.

Reviewer 4 response

Modeling concerns:

To better clarify the differences between steady-state and non-steady-state cores in this manuscript, we have decided to model all cores using the same setup without fitting porewater sulfate data. (This is the reduced model setup shown only for non-steady-state cores in the previous version of the manuscript). Large differences in the run time to match observed profiles (Figure 2) allow us to justify categorizing these cores as steady-state (Fig. 2A-D) vs non-steady-state (Fig. 2I-L). Please note that we are now considering GC1048 as a third transitional state due to the slight bend in its porewater sulfate profile below 2.5 m (Fig. 2G). This fits a distinct modeling scenario of an increasing (though much lower) methane flux, and a SMT shoaling speed of 0.4 cm/yr instead of 10. As such, we have placed porewater, community, and gene abundance data from this core separately in supplemental figure S3, and omitted communities from this core in subsequent analyses which depend on steady-state vs non-steady-state classifications. In the Results subsection "Identification of distinct states of methane transport" (lines 143-179), we more clearly describe how our modeling delineates the classification of cores by states of methane dynamics (please note that we have referred to these as "regimes" in the previous version of the manuscript). We address major assumptions in the figures and text of the supplementals.

The modeling results, or the classification of steady-state, transitional, and non-steady-state, is not only based on the observations presented in this work. Rather, such a classification reflects the accumulation of knowledge from previous work over the past five years, some of which we are involved in. We have listed all sediment cores from Storfjordrenna that have been collected and studied, and summarized some of the previous work on geochemistry, biology, and microbiology in footnotes of Table S3. In addition, based on geophysical evidence, Waage et al. (2019, Geochemistry, Geophysics, Geosystems) have observed contrasting fluid migration behaviors beneath the GHMs studied here that supports our classification of the different states. In summary, our model results (presented as hypothetical cases in the revised manuscript, lines 149-153) are firmly supported by independent observations and serve as a strong foundation for our analysis of microbial community diversity, abundance, and dynamics at these GHMs.

Assuming sulfate reduction is all attributed to anaerobic methane oxidation:

While we concede that not 100% of the sulfate is reduced by AOM at these sites, porewater ammonium data from cores analyzed in this study and from some of our previous work at Storfjordrenna reveal that methane is a much larger sink for sulfate than organic matter (Fig. S2, and additional maximum porewater ammonium concentrations from all available cores at Storfjordrenna are provided in Supplementary Table 3). AOM-independent sulfate reducing lineages, if active, may be responsible for a small fraction of organic matter-dependent sulfate reduction. We argue that this is negligible in the context of our modeling setup, because it is adequate to reproduce shapes of sulfate porewater profiles in steady-state and non-steady-state cores (revised Fig. 2).

Lack of FISH, metagenomic data:

We appreciate the reviewer's suggestion that FISH would directly reveal associations between ANME and SRB to verify coupled AOM/SR. Gründger et al (2019) used FISH to document associations between ANME-1b and SEEP-SRB1 in a biofilm recovered from GC1048, one of the cores discussed in this study, though we do not have FISH data for additional samples. To investigate potential interactions with 16S data, we have constructed a co-occurrence network of ASVs from all microbial communities in this study (Figure S8). Though this limits us to inferring *potential* interactions, this approach has been recently used to identify co-occurrences between ANME and SRB that were experimentally validated with FISH (Metcalf et al, 2020, ISME) as well as ANME co-occurrences with other community members (Niu et al, 2017, FEMS Microbiology Ecology). We identify high co-occurrences between many abundant ANME and SRB, though we note a few interactions that could be indicative of organic matter decomposition coupled with sulfate reduction. We did not perform any metagenomic sequencing on these samples that could have led to a gene- or MAG-centric characterization of potential metabolisms or interactions within the community.

Gene quantification:

RNA recovery from sediments is challenging, and timescales of mRNA degradation and core recovery (half an hour or more) present barriers to reliably obtaining and quantifying transcripts of functional genes. While we concede that quantification of *mcrA* genes from extracted DNA

does not directly indicate active communities undergoing AOM, we still found it notable that they varied several orders of magnitude across cores experiencing increases in methane flux and peaked at sediment depths corresponding to maximum numerically-derived AOM rates. That our estimates of ANME doubling times and cell-specific AOM rates fall within other values reported by others (Ruff et al., ISME, 2019 and Girguis et al., Applied & Environmental Microbiology, 2005, lines 446-449) gives us confidence that our estimates may be reasonable. Though, to eliminate the possibility of misunderstanding, we add that we do not have direct measures of cell activities (line 443).

Advective transport of microbial aggregates:

Steady-state and non-steady-state phases described in this manuscript are diffusion-dominated systems, where we do not expect any fluid or aggregate transport. We relied particularly on increases in *mcrA* gene numbers and relative abundances of ANME in the two non-steady-state cores (GC1045 and GC1081) to infer microbial community changes to rapid influxes of methane. We do not argue these community changes in PC1029, the core from the active seep. Even so, the active seep system is likely to be dominated by decoupling of fluids so that methane in the gas phase advects separately from the aqueous phases, as Hong et al., 2018 (Geophysical Research Letters) show in the Storfjordrenna area (lines 421-423).

What we did (to describe to editor)

As per the suggestions of Reviewer 2, we have decided to model AOM rates from hypothetical profiles of sulfate and methane instead of profiles fitted to the data. This approach is what we presented in the previous version of the manuscript as the “reduced model” for non-steady-state cores; this is the only modeling we present in the updated version. In non-steady and steady-state cores, we see large differences in model run times that produce sulfate profiles that correspond closely with our data (now Figure 2). This gives us additional confidence in our classification scheme that we elaborate on in the rest of the manuscript. This also brought to our attention that core GC1048 did not fit either steady or non-steady state classification, so we omitted microbial communities from this core from the main manuscript. Please note that in addition to adding a figure showing differences in model output between steady-state and non-steady-state cores (Figure 2), we have removed AOM rate profiles from steady-state cores (Figure 3) and removed descriptions and figures showing the fitted modeling approach from the supplementals. Modeled AOM rates from steady-state cores in Fig. S5 are based on updated rate calculations.

We have also made a few other changes to figures in the supplemental information. Whole-core methane flux calculations, previously Fig. S1, are now in Table S1. Figure 5, which showed differentially abundant ASVs across states of methane dynamics, has been moved to the supplementals (Fig. S6). We have addressed concerns Reviewer 4 had about our assumption that all sulfate was reduced by AOM in Figure S2, where modest ammonium concentrations downcore point towards minimal organoclastic sulfate reduction. This is in line with other cores

collected at Storfjordrenna; we have added a supplementary table with maximum porewater ammonium concentrations from all such studies (Table S3). Reviewer 4 also asserted that FISH microscopy data was mandatory to claim interactions between ANME and SRB. While we disagree with this viewpoint because these interactions are supported by dozens of studies over two decades, we did not collect any FISH samples. Instead, we were able to construct a co-occurrence network to show that there are several highly-abundant ANME/SRB whose percent abundances across all communities in this study coincide highly. This network approach has been recently used to identify co-occurrences between ANME and SRB that were experimentally validated with FISH (Metcalf et al, 2020, ISME).

REVIEWERS' COMMENTS

Reviewer #2 (Remarks to the Author):

Review Klasek et al, Distinct methane-dependent biogeochemical states in Arctic seafloor gas hydrate mounds

With apologies for the delay in this review, the summer is never a great time to agree to review a paper. This is the third or fourth time I have seen this paper and I appreciate the authors' consideration and addressing my comments on the previous versions. I think the paper is ready to be accepted, and include my final comments below.

The contribution of this study, and why I think it belongs in Nature Comms is that it combines geomicrobiological and geochemical/reactive transport modelling and shows how the inorganic geochemistry can be linked to changes in microbial communities and composition/diversity. There are very very few studies that link these together and on this alone it should be published.

Larger comment:

- Much of the paper is about the deviation from steady state of the sulfate concentration porefluid profiles. But this should be specified that what the authors mean is the concave up nature of the pore fluid profiles. Steady state with constant consumption of sulfate with depth would also result in a deviation from a linear pore fluid profile, with a concave down concentration gradient. This needs to be clarified in the introduction (Lines 85-95) and elsewhere. It is also fair to point out that when you have two points making a line they will make... a line. You don't know if this is concave up or concave down in the absence of other data points. This lack of data needs to be clarified (Line 155-160)

Smaller comments:

Line 26 – I still find 'methane flux setting' to be awkwardly phrased. Can you find another way to express this? Even 'distinct methane concentration profiles'

Line 44 – and end up in the overlying water column – I'd add this.

Line 77 – 'this habitat' is vague, can you remind the reader what you mean?

Line 85 – add 'concentrations' to after 'sulfate'

Line 88 – I'd specify that the deviation to linearity is the concave UP nature of the profiles – see above in the 'larger comments' section.

Line 121-122 – I am struggling with the word 'contrast' here. I read it the first time as the alkalinity profiles are inconsistent with the sulfate profiles -that is if you were to plot them one a cross plot you would find a deviation from the 1:1 line that would be expected for sulfate-coupled AOM. But I think what you mean is just that alkalinity increases while sulfate decreases. Could this be said more simply?

Line 148 – here you use low ammonium concentrations to argue for no organoclastic sulfate reduction...

Line 152 – approaches steady-state (not 'a steady-state condition')

Line 196 – I don't like this title – Steady-state doesn't necessarily mean biogeochemical equilibrium.

Line 254 – I think there is a typo (lifestyles)

Line 412 – 'the upward flux of methane' rather than 'methane fluxes'.

Line 412 – elsewhere, please try not to start sentences with acronyms. Rates of AOM is better anyway to kick off this sentence.

Line 421-423 – This sentence is very awkwardly phrased, I had to read it several times and still couldn't get my head around it.

Line 428 – 'flux values' is awkward. The magnitude of the methane flux?

Reviewer #4 (Remarks to the Author):

In the revised version of the manuscript, authors addressed some of my previous comments. My major concern at this point is that I am having hard time to determine what are the new results of this research. Influence of fluid fluxes on microbial community, abundance and activity, notably AOM and SR microorganisms, is not new and several papers already documented this relationship. Authors might consider to better highlight the novelty of their findings. For instance, in the Summary section, none of the A) B) or C) is a new result. Wrapping the dataset in a mathematical model was a good idea but the benefit in term of microbial ecology understanding sounds low and adds in my opinion only little value to the actual knowledge in seep microbiology.

In addition, I am still doubtful regarding the validity of the model and the assumption that SR is only associated with methane oxidation. Ammonium profiles doesn't exclude the possibility of SR linked to hydrocarbons (short alkanes) degradation. Is there any other hydrocarbon beside methane in the seepage? The region is well known to be rich in oil and gas reservoirs. Can the authors exclude this hypothesis?

Reviewer #2 (Remarks to the Author):

Review Klasek et al, Distinct methane-dependent biogeochemical states in Arctic seafloor gas hydrate mounds

With apologies for the delay in this review, the summer is never a great time to agree to review a paper. This is the third or fourth time I have seen this paper and I appreciate the authors' consideration and addressing my comments on the previous versions. I think the paper is ready to be accepted, and include my final comments below.

The contribution of this study, and why I think it belongs in Nature Comms is that it combines geomicrobiological and geochemical/reactive transport modelling and shows how the inorganic geochemistry can be linked to changes in microbial communities and composition/diversity. There are very very few studies that link these together and on this alone it should be published.

Larger comment:

- Much of the paper is about the deviation from steady state of the sulfate concentration porefluid profiles. But this should be specified that what the authors mean is the concave up nature of the pore fluid profiles. Steady state with constant consumption of sulfate with depth would also result in a deviation from a linear pore fluid profile, with a concave down concentration gradient. This needs to be clarified in the introduction (Lines 85-95) and elsewhere. It is also fair to point out that when you have two points making a line they will make... a line. You don't know if this is concave up or concave down in the absence of other data points. This lack of data needs to be clarified (Line 155-160)

To clarify, we are assuming a linear sulfate profile as an initial condition, with all SR coupled to AOM at the SMT (instead of a concave-down shape as described above). We have addressed this in lines 83-85 (and later discuss the validity of assuming nearly all SR coupled to AOM).

We have amended the sentence discussing cores with linear decreases in sulfate concentration: "though the sparsity of sulfate concentration data for these cores adds some uncertainty to this interpretation" (lines 158-160). We have also changed descriptions of "linear" to "approximately linear" (line 199).

Smaller comments:

Line 26 – I still find 'methane flux setting' to be awkwardly phrased. Can you find another way to express this? Even 'distinct methane concentration profiles'

Suggestion accepted, line 26.

Line 44 – and end up in the overlying water column – I'd add this.

Suggestion accepted, line 44.

Line 77 – 'this habitat' is vague, can you remind the reader what you mean?

Changed "this habitat" to "methane-rich marine sediments", line 77. Also rephrased "unknown" to "poorly-understood", as the Ruff et al ISME paper (25) came out after our first draft and really examines this question in a different methane-rich seafloor setting.

Line 85 – add 'concentrations' to after 'sulfate'

This sentence has been reworded (lines 83-85).

Line 88 – I'd specify that the deviation to linearity is the concave UP nature of the profiles – see above in the 'larger comments' section.

As described above, we are initially assuming linearity, though we have clarified here that this deviation is a transition from a linear to concave up shape (line 87).

Line 121-122 – I am struggling with the word 'contrast' here. I read it the first time as the alkalinity profiles are inconsistent with the sulfate profiles -that is if you were to plot them one a cross plot you would find a deviation from the 1:1 line that would be expected for sulfate-coupled AOM. But I think what you mean is just that alkalinity increases while sulfate decreases. Could this be said more simply?

Changed to "All cores show downcore increases in alkalinity throughout the sulfate reduction zone" (lines 120-121).

Line 148 – here you use low ammonium concentrations to argue for no organoclastic sulfate reduction...

Please see lines 9-21 of the supplemental information, Figure S2, and Table S3 for our justification of this.

Line 152 – approaches steady-state (not 'a steady-state condition')

Suggestion accepted, line 151.

Line 196 – I don't like this title – Steady-state doesn't necessarily mean biogeochemical equilibrium.

We have changed this to "steady-state pore fluid system" (line 197). At the editor's request, we have also removed secondary subheadings (each methane state is now its own subheading, lines 220 and 255 as well).

Line 254 – I think there is a typo (lifestyles)

Good catch, line 257.

Line 412 – 'the upward flux of methane' rather than 'methane fluxes'.

Suggestion accepted, line 414.

Line 412 – elsewhere, please try not to start sentences with acronyms. Rates of AOM is better anyway to kick off this sentence.

Suggestion accepted, line 415 (and similarly in line 409).

Line 421-423 – This sentence is very awkwardly phrased, I had to read it several times and still couldn't get my head around it.

Rephrased to "Though gas advects during seepage, solute transport in surrounding porewaters may remain governed by diffusion, as this decoupling of fluid transport mechanisms has been described at Storfjordrenna methane seeps" (lines 424-427).

Line 428 – 'flux values' is awkward. The magnitude of the methane flux?

Suggestion accepted, line 432.

Reviewer #4 (Remarks to the Author):

In the revised version of the manuscript, authors addressed some of my previous comments. My major concern at this point is that I am having hard time to determine what are the new results of this research. Influence of fluid fluxes on microbial community, abundance and activity, notably AOM and SR microorganisms, is not new and several papers already documented this relationship. Authors might consider to better highlight the novelty of their findings. For instance, in the Summary section, none of the A) B) or C) is a new result. Wrapping the dataset in a mathematical model was a good idea but the benefit in term of microbial ecology understanding sounds low and adds in my opinion only little value to the actual knowledge in seep microbiology.

How quickly microbial communities change, and the populations of ANME and SRB establish after methane influx in situ, is a question currently addressed by only one other study to our knowledge (Ruff et al 2018, ISME, citation 25). That study examined a different setting (Haakon Mosby Mud Volcano) than Storfjordrenna, which is characterized by diffusive solute transport and high methane fluxes relative to other seep systems. In particular, the paragraph beginning at line 446 discusses how quickly ANME populations increase in response to methane influx, which is impossible to constrain without a modeling approach or extremely high-resolution seafloor sampling and rate measurements. We more explicitly emphasize the novelty of these findings in lines 461-463.

In addition, I am still doubtful regarding the validity of the model and the assumption that SR is only associated with methane oxidation. Ammonium profiles doesn't exclude the possibility of SR linked to hydrocarbons (short alkanes) degradation. Is there any other hydrocarbon beside methane in the seepage? The region is well known to be rich in oil and gas reservoirs. Can the authors exclude this hypothesis?

In the seeping fluid sampled at the seafloor, $C1/\sum C2-C5$ is greater than 1300 (Hong et al., 2018, citation 32), which means there is only less than 76 vppm of ethane to pentane altogether. Higher concentrations of ethane and propane were documented by Serov et al. (2017, citation 17) for the dissolved gas in sediment pore fluid ($C1/(C2+C3)=200-656$, $n=12$, for the samples below sulfate reduction zone) that is between 13 to 57 vppm. We believe such a trace amount of hydrocarbon has very little effect on the overall sulfate reduction.